# GRLA: Bridging Softmax and Linear Attention via Gaussian RBF Kernel for Lightweight Image Super-Resolution

## Abstract

Lightweight image super-resolution (SR) requires effective modeling of long-range dependencies under stringent computational constraints. Although self-attention mechanisms are highly effective for this task, their quadratic computational complexity imposes a prohibitive constraint in lightweight SR applications. Existing linear attention methods reduce complexity to linear but significantly underperform compared to Softmax attention due to their inability to explicitly model the Euclidean distance between query and key vectors. Through mathematical derivation, we demonstrate that the core operation of standard Softmax attention, $\exp(Q_i^T K_j)$, is equivalent to an unnormalized Gaussian Radial Basis Function (GRBF) kernel. Building on this insight, we propose a GRBF-based linear attention mechanism (GRBFLA), which reformulates a distance-aware GRBF kernel that is amenable to Taylor series expansion, enabling linear approximation. This kernel progressively approximates the behavior of standard Softmax attention while maintaining linear complexity. Based on GRBFLA, we develop a lightweight image SR architecture termed GRLA. Experimental results show that for ×4 SR on the Manga109 dataset, GRLA outperforms the representative self-attention model SwinIR-light by 0.57 dB in PSNR while reducing computational cost FLOPs by 11%. Compared to the state-of-the-art Mamba-based lightweight model MambaIRv2-light, GRLA achieves a 0.25 dB higher PSNR with a 25% reduction in FLOPs.

## 1 Introduction

Image super-resolution (SR) (Dong et al., 2014; Timofte et al., 2016), a core task in computer vision, aims to reconstruct high-resolution (HR) images from low-resolution (LR) inputs. It has broad applications in medical image enhancement (Sarkar et al., 2022; Chaudhari et al., 2018), satellite remote sensing (Jiang et al., 2019), and boosts downstream tasks like object detection (Hsu & Chen, 2022) and semantic segmentation (Tian et al., 2022), as its reconstruction quality directly impacts subsequent analysis accuracy. In resource-constrained edge scenarios, lightweight SR models need to balance compactness with strong long-range dependency modeling and high-frequency detail recovery capabilities. Thus, designing an efficient linear-complexity mechanism for long-range dependency modeling remains a key challenge in lightweight image SR.

Deep learning-based super-resolution (SR) methods, especially those using convolutional neural networks (CNNs) (Huang et al., 2015; Ledig et al., 2017; Lim et al., 2017; Qiu et al., 2019; Rad et al., 2019; Song et al., 2021), have advanced significantly by learning end-to-end LR-to-HR mapping. However, they have inherent limitations: conventional convolutional layers cannot adaptively model pixel-wise dependencies, and expanding receptive fields requires stacking layers (increasing depth and computation). While lightweight CNN models (Ahn et al., 2018; Hui et al., 2018; 2019; Li et al., 2020; Liu et al., 2020; Luo et al., 2020) reduce model size and complexity, their local receptive fields still limit long-range dependency capture.

To address these inherent limitations, the self-attention (SA) mechanism derived from Transformers (Vaswani et al., 2017) has been incorporated into SR models, enabling the modeling of dependencies between distant image regions. SA computes similarity weights between all pixel pairs, explicitly

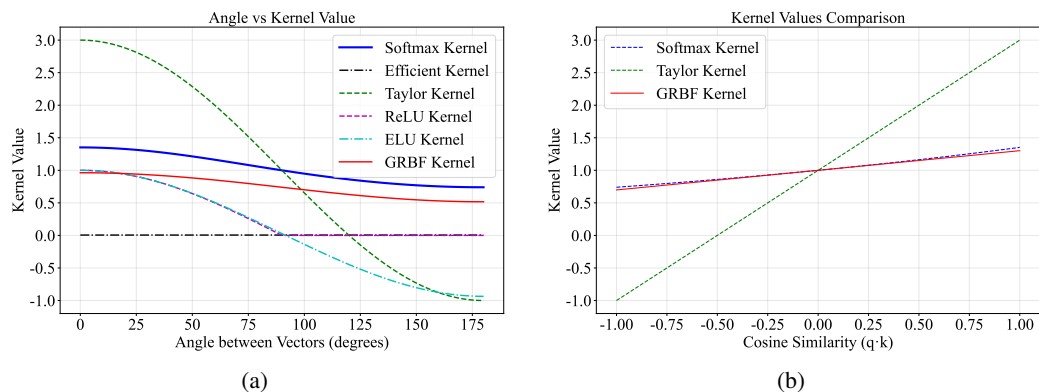

Figure 1: (a) Comparison of kernel values versus vector angles for different kernels. The GRBF kernel exhibits similar characteristics to the Softmax kernel, while maintaining effective distance awareness. (b) Comparison of kernel values: the Taylor-approximated GRBF kernel closely matches the kernel values of the Softmax kernel.

establishing long-range dependencies. However, the quadratic computational complexity of SA restricts its practical applicability for large-scale images. This bottleneck has driven the development of efficient attention variants, including SwinIR-light (Liang et al., 2021), ELAN (Zhang et al., 2022), SRformer-light (Zhou et al., 2023), Restormer (Zamir et al., 2022), and DCTLSA (Zeng et al., 2023), which reduce the computational overhead of Softmax attention or even achieve linear computational complexity. Nonetheless, these methods often sacrifice the capability of long-range dependency modeling for improved efficiency, resulting in suboptimal high-frequency detail reconstruction.

In contrast to modifications to Softmax attention, kernel-based linear attention fundamentally restructures the computational process of SA. Linear attention eliminates the Softmax operation and approximates the original $\exp(Q_i^T K_j)$ term, thereby achieving linear computational complexity. However, theoretical analysis indicates that simple mapping functions for constructing similarity kernels fail to effectively approximate the distance-aware characteristics of standard Softmax attention. A breakthrough in this research direction would provide crucial theoretical support for the development of efficient Transformer-based architectures.

To narrow the performance gap induced by the limited expressiveness of existing linear attention kernels, we propose Gaussian Radial Basis Function (GRBF)-based linear attention (GRBFLA). This method employs the GRBF kernel as a similarity metric, explicitly and directly quantifying similarity via exponential decay based on Euclidean distance. Theoretical analysis indicates that the core computation of standard Softmax attention is equivalent to an unnormalized GRBF kernel, which thereby reveals that Softmax attention is inherently distance-aware. Fig 1(a) shows the comparison between the GRBF kernel and other representative kernels (Shen et al., 2021; Qiu et al., 2023; Cai et al., 2023; Fan et al., 2025). The GRBF kernel closely mimics the characteristics of Softmax attention, with its values decreasing as the angle between vectors increases. We adapt this inherently distance-aware GRBF kernel to the linear attention framework by decomposing the squared Euclidean distance and applying a first-order Taylor approximation to the exponential inner product term. This reformulation yields a linearly computable form that preserves distance awareness and progressively converges to standard Softmax attention as $\gamma \to 0$. As illustrated in Fig. 1(b), this linear approximation exhibits high accuracy within the principal operating region of SR models. Based on GRBFLA, we further propose GRLA, a method that outperforms other linear attention methods. The overall network structure is detailed in Appendix 3.4.

To validate the effectiveness of GRBFLA, we employ Local Attribution Map (LAM) (Gu & Dong, 2021)-based visualizations to compare it with several representative linear attention methods (Shen et al., 2021; Qiu et al., 2023; Cai et al., 2023; Fan et al., 2025). As illustrated in Fig. 2, GRBFLA generates wider attribution regions and higher Diffusion Index (DI) values, which in turn activate more pixels and leverage richer contextual information to achieve higher-quality SR reconstruction

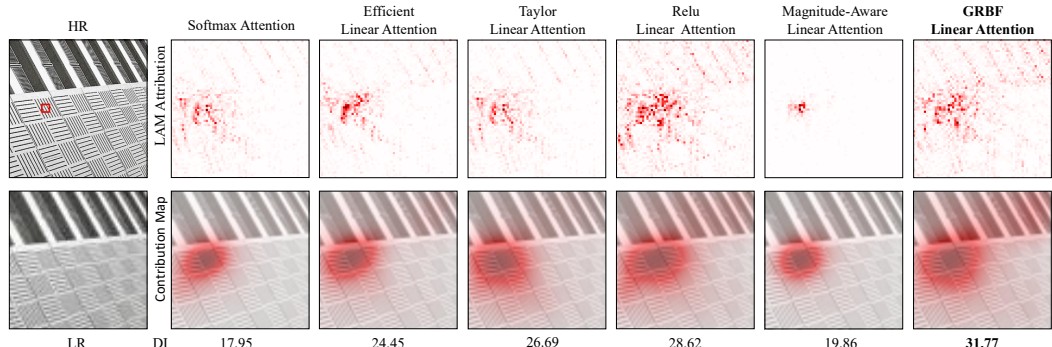

Figure 2: Local Attribution Map (LAM) (Gu & Dong, 2021)-based visualizations of different attention methods. The Diffusion Index (DI) reflects the extent of involved pixels, where a higher DI indicates broader pixel utilization for SR reconstruction.

results. This demonstrates that GRBFLA can effectively capture Euclidean distance-sensitive long-range dependencies and exhibits strong spatial dependency modeling capabilities.

The main contributions of this work are summarized as follows:

1. We propose leveraging the Gaussian Radial Basis Function (GRBF) kernel as the foundation for similarity measurement in self-attention. Via mathematical derivation, we adapt and reformulate it into a form compatible with linear attention decomposition, while preserving inherent distance awareness.

2. Under first-order Taylor approximation (i.e., as $\gamma \rightarrow 0$), the reformulated GRBF kernel progressively converges to the core computation of standard Softmax attention. This not only reduces the computational complexity from quadratic to linear but also ensures its ability to model long-range dependencies.

3. This work bridges the performance gap between long-range dependency modeling and lightweight design, offering a new paradigm for efficient image SR. Additionally, the proposed design can be readily integrated into existing CNN and Transformer-based architectures, showing broad applicability.

## 2 RELATED WORK

### 2.1 CNN-BASED METHODS

With the advancement of deep learning, convolutional neural network (CNN)-based super-resolution (SR) methods have achieved remarkable success. SRCNN (Dong et al., 2014) employs a three-layer convolutional architecture to directly learn an end-to-end mapping relationship from low-resolution (LR) to high-resolution (HR) images. Recent lightweight SR methods include CARN (Ahn et al., 2018), which combines residual and recursive learning; IDN (Hui et al., 2018), which uses channel splitting to create compact information distillation blocks; IMDN (Hui et al., 2019), which introduces incremental multi-distillation blocks; RFDN (Liu et al., 2020), which proposes simplified residual blocks with feature distillation connections; and LatticeNet (Luo et al., 2020), which combines multiple residual blocks in a butterfly structure along with reverse feature fusion. Although significant progress has been made in lightweight SR research, there remains room for improvement in the performance of lightweight SR models.

### 2.2 TRANSFORMER-BASED METHODS

Transformers (Vaswani et al., 2017), originally designed for natural language processing (NLP), have been widely applied to various deep learning tasks. Recently, self-attention (SA) mechanisms have been adopted in low-level computer vision tasks. SwinIR-light (Liang et al., 2021), based on the Swin Transformer (Liu et al., 2021), employs a shifted window scheme to compute SA within

small non-overlapping windows, which indirectly learns long-range dependencies via cross-window aggregation. ELAN (Zhang et al., 2022) proposes an efficient long-range attention mechanism that uses shared attention to reduce model parameters, which in turn forms a lightweight SR model. SRFormer-light (Zhou et al., 2023) proposes a novel SR-oriented permuted self-attention method. These methods leverage SA to capture long-range dependencies between image regions, which in turn aids high-frequency detail reconstruction in SR tasks. However, the quadratic computational complexity of SA makes it challenging to process HR images, which in turn limits its practical applicability in lightweight models.

## 2.3 LINEAR ATTENTION METHODS

Linear attention reduces computational complexity to linear order via kernel product factorization but sacrifices performance by lacking explicit modeling of query-key Euclidean distance, crucial for spatial structural dependencies. For instance, Restormer (Zamir et al., 2022) and DCTLSA (Zeng et al., 2023) apply self-attention along channels instead of spatially, cutting complexity but losing useful spatial information for SR. Recent Mamba architecture shows potential in modeling long-range dependencies with linear complexity. MambaIR-light (Guo et al., 2024) applies Mamba (Gu & Dao, 2023) to low-level vision using causal scan blocks, with MambaIRv2-light (Guo et al., 2025) optimizing scanning order for better restoration. However, Mamba's state space model differs fundamentally from similarity-weighted attention, making it hard to approximate Softmax attention, while its scanning mechanism introduces unnatural sequential assumptions for images and high overhead. In contrast, this work reveals the mathematical equivalence between Gaussian Radial Basis Function kernel and Softmax attention's core computation, constructing an $O(n)$ linear attention architecture. It addresses existing linear attention's performance degradation from poor distance awareness and applies it to lightweight image SR.

## 3 METHOD

### 3.1 REVISITING LINEAR ATTENTION

The self-attention mechanism in Transformers operates as follows: given an input feature map $X \in R^{H \times W \times C}$, where $H$, $W$, and $C$ denote the height, width, and number of channels, three learnable projection matrices $W_Q$, $W_K$, and $W_V$ are employed to generate query vectors $Q = XW_Q$, key vectors $K = XW_K$, and value vectors $V = XW_V$. Self-attention score is then computed as:

$$\alpha_i = \sum_{j=1}^{N} \frac{\text{Sim}(Q_i, K_j)}{\sum_{j=1}^{N} \text{Sim}(Q_i, K_j)} V_j \qquad (1)$$

where $\text{Sim}(\cdot)$ denotes a similarity measurement function. In standard Softmax attention, $\text{Sim}(Q_i, K_j) = \exp(Q_i^T K_j)$ (the scaling factor is omitted for simplicity). This computation requires calculating exponentials for all query-key ($Q$-$K$) pairs, leading to a time complexity of $O(n^2)$ complexity (where $n = H \times W$ denotes the total number of spatial tokens in the input feature map). Linear attention designs a kernel function $\phi(\cdot)$ to approximate the aforementioned similarity measurement function and maps $Q$ and $K$ to a positive real-valued space such that $\text{Sim}(Q_i, K_j) = \phi(Q_i)^T \phi(K_j)$. Based on this approximation, linear attention computation can be reformulated as follows:

$$\alpha_i = \sum_{j=1}^{N} \frac{\phi(Q_i)^T \phi(K_j)}{\sum_{j=1}^{N} \phi(Q_i)^T \phi(K_j)} V_j = \frac{\phi(Q_i)^T \sum_{j=1}^{N} \phi(K_j) V_j}{\phi(Q_i)^T \sum_{j=1}^{N} \phi(K_j)} \qquad (2)$$

This reformulated form circumvents the explicit computation of pairwise similarity scores and reduces the time complexity to $O(n)$. However, kernel functions structured as $\text{Sim}(Q_i, K_j) = \phi(Q_i)^T \phi(K_j)$ often fail to adequately express or approximate the complex nonlinear similarity relationships based on vector distances inherent in standard Softmax attention, particularly its distance sensitivity. This results in weaker long-range dependency modeling capabilities compared to standard Softmax attention, which is the root cause of performance degradation in existing linear attention methods.

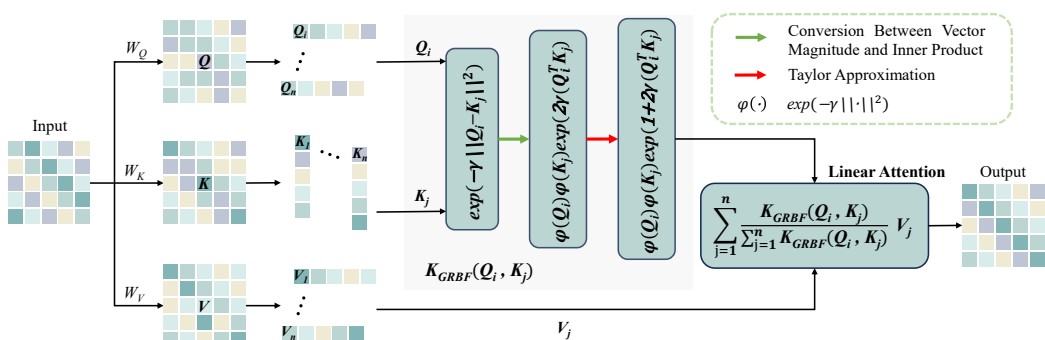

Figure 3: Gaussian Radial Basis Functions Linear Attention.

## 3.2 GAUSSIAN RADIAL BASIS FUNCTION

Addressing the insufficient distance awareness in existing linear attention kernel functions, we employ the Gaussian Radial Basis Function (GRBF) kernel as the similarity metric. The GRBF kernel $\exp(-\gamma\|Q_i - K_j\|^2)$ naturally and explicitly measures similarity between vectors through exponential decay based on Euclidean distance. Notably, we mathematically derive that the core computation of standard Softmax attention, $\exp(Q_i^T K_j)$, is equivalent to an unnormalized GRBF kernel under the consideration of vector norms. This key finding demonstrates that the GRBF kernel serves as a more fundamental and direct choice for constructing high-performance linear attention mechanisms, which effectively addresses the distance awareness deficiency of existing kernels discussed earlier. The core objective of this work is to adapt this highly expressive GRBF kernel to the linear attention computational framework, which thereby enables linear-complexity attention with inherent distance awareness.

As illustrated in Fig. 3, let $Q_i$ and $K_j$ denote two vectors in the input feature map. Their GRBF-based similarity kernel function is formally defined as follows:

$$K_{\text{GRBF}}(Q_i, K_j) = \exp\left(-\gamma\|Q_i - K_j\|^2\right) \tag{3}$$

where $\exp(\cdot)$ is the exponential function, $\gamma > 0$ is the bandwidth parameter controlling the influence of distance on similarity, and $\|Q_i - K_j\|^2$ is the squared Euclidean distance, reflecting the difference between the two vectors. A smaller value indicates closer proximity, providing explicit distance awareness. Additionally, the GRBF kernel incorporates nonlinear activation, which implicitly enhances the feature representational capacity and generalization performance of the SR model.

However, the squared Euclidean distance term in Eq. (3) necessitates specific mathematical treatment to adapt it to the linear attention computational framework. Considering that matrix multiplication in self-attention can be interpreted as an extension of inner products between row vectors of the query matrix and column vectors of the key matrix, we decompose the squared Euclidean distance term using the well-established mathematical relationship between vector norms and inner products, as follows:

$$\|Q_i - K_j\|^2 = \|Q_i\|^2 + \|K_j\|^2 - 2Q_i^T K_j \tag{4}$$

where $\|Q_i\|^2$ and $\|K_j\|^2$ are the squared L2 norms of $Q_i$ and $K_j$, respectively, reflecting their magnitudes. $Q_i^T K_j$ is the inner product of the two vectors. Substituting the decomposed squared Euclidean distance term into the GRBF kernel definition (Eq. 3) yields the following reformulated GRBF kernel expression:

$$K_{\text{GRBF}}(Q_i, K_j) = \exp\left(-\gamma\left(\|Q_i\|^2 + \|K_j\|^2 - 2Q_i^T K_j\right)\right) \tag{5}$$

Splitting the exponential term in the reformulated GRBF kernel expression into product terms yields the following expression:

$$K_{\text{GRBF}}(Q_i, K_j) = \exp\left(-\gamma\|Q_i\|^2\right)\exp\left(-\gamma\|K_j\|^2\right)\exp\left(2\gamma Q_i^T K_j\right) \tag{6}$$

where $\exp(-\gamma\|Q_i\|^2)$ and $\exp(-\gamma\|K_j\|^2)$ are exponential terms of the vector norms, which can be viewed as weightings of the vectors' own importance. This reformulated expression establishes

an explicit mathematical relationship between the GRBF kernel, the squared L2 norms of $Q$-$K$ vectors, and the $Q$-$K$ inner product, which thereby provides a solid theoretical foundation for the subsequent linear approximation of the GRBF kernel. For simplicity, let $\varphi(Q_i) = \exp(-\gamma||Q_i||^2)$ and $\varphi(K_j) = \exp(-\gamma||K_j||^2)$. Then:

$$K_{\text{GRBF}}(Q_i, K_j) = \varphi(Q_i)\varphi(K_j) \exp\left(2\gamma Q_i^T K_j\right) \tag{7}$$

Thus, the standard GRBF kernel $\exp(-\gamma||Q_i - K_j||^2)$ can be decomposed into norm terms $\varphi(Q_i)$, $\varphi(K_j)$, and an exponential inner product term $\exp(2\gamma Q_i^T K_j)$.

### 3.3 First-order Taylor approximation

However, the exponential inner product term $\exp(2\gamma Q_i^T K_j)$ in Eq. (7) hinders the kernel function $\text{Sim}(Q_i, K_j)$ from being decomposed into the form $\phi(Q_i)^T \phi(K_j)$, which is key to achieving linear computational complexity. To adapt the GRBF kernel to the linear attention computational framework while preserving its inherent distance-aware properties, we introduce a first-order Taylor approximation for the exponential inner product term. When $2\gamma Q_i^T K_j$ is small (achieved by L2 normalization of $Q_i$ and $K_j$ and setting a small bandwidth parameter $\gamma$), this approximation is sufficiently accurate. As illustrated in Fig. 1(b), the Taylor approximation of the GRBF kernel achieves a more accurate approximation of standard Softmax attention compared to existing simple Taylor approximation (Qiu et al., 2023). In practical experiments, we find that setting the bandwidth parameter $\gamma = 1/2(\times\sqrt{d})$ (see Appendix A.3 for details) yields the optimal SR reconstruction results. Based on this, we approximate the exponential inner product term as:

$$\exp\left(2\gamma Q_i^T K_j\right) \approx 1 + 2\gamma Q_i^T K_j \tag{8}$$

This first-order Taylor approximation has a solid mathematical foundation, transforming the nonlinear exponential term into a decomposable linear form. Substituting the Taylor-approximated exponential inner product term (Eq. 8) into the decomposed GRBF kernel expression (Eq. 7), we obtain a decomposable approximate GRBF kernel function tailored to the linear attention computational framework:

$$K_{\text{GRBF}}(Q_i, K_j) \approx \varphi(Q_i)\varphi(K_j)\left(1 + 2\gamma Q_i^T K_j\right) \tag{9}$$

where $\varphi(Q_i)$ and $\varphi(K_j)$ retain norm information, preserving the distance-aware properties of the original GRBF kernel. Substituting the above-derived decomposable approximate GRBF kernel into the general linear attention computation formula (Eq. 2), we derive the output expression for the Gaussian Radial Basis Function (GRBF)-based linear attention (GRBFLA):

$$\alpha_i = \frac{\sum_{j=1}^{N} \varphi(Q_i)\varphi(K_j)\left(1 + 2\gamma Q_i^T K_j\right) V_j}{\sum_{j=1}^{N} \varphi(Q_i)\varphi(K_j)\left(1 + 2\gamma Q_i^T K_j\right)} \tag{10}$$

Further simplification yields:

$$\alpha_i = \frac{\varphi(Q_i) \sum_{j=1}^{N} \varphi(K_j)\left(1 + 2\gamma Q_i^T K_j\right) V_j}{\varphi(Q_i) \sum_{j=1}^{N} \varphi(K_j)\left(1 + 2\gamma Q_i^T K_j\right)} \tag{11}$$

The norm-related term $\varphi(Q_i)$ acts as a common factor in both the numerator and denominator of the GRBFLA output expression. Thus, it can be mathematically canceled out without affecting the relative attention weights. In contrast, the norm-related term $\varphi(K_j)$ is explicitly retained in the expression, and this retention is critical to preserving the distance-aware property of the GRBFLA, as $\varphi(K_j)$ encodes the squared L2 norm information of $(K_j)$. After the above cancellation and retention operations, the final linearly computable attention output for GRBFLA, denoted as $\alpha_i$, is given by the following formula:

$$\alpha_i = \frac{\sum_{j=1}^{N} \varphi(K_j)V_j + 2\gamma Q_i^T \sum_{j=1}^{N} \varphi(K_j)K_jV_j}{\sum_{j=1}^{N} \varphi(K_j) + 2\gamma Q_i^T \sum_{j=1}^{N} \varphi(K_j)K_j} \tag{12}$$

where $\sum_{j=1}^{N} \varphi(K_j)V_j$ denotes the value vector sum weighted by the norm-related terms of key vectors, containing global context information, as it aggregates value vectors $V_j$ across all spatial positions via key-based weighting. $\sum_{j=1}^{N} \varphi(K_j)K_jV_j$ represents the key-value interaction sum

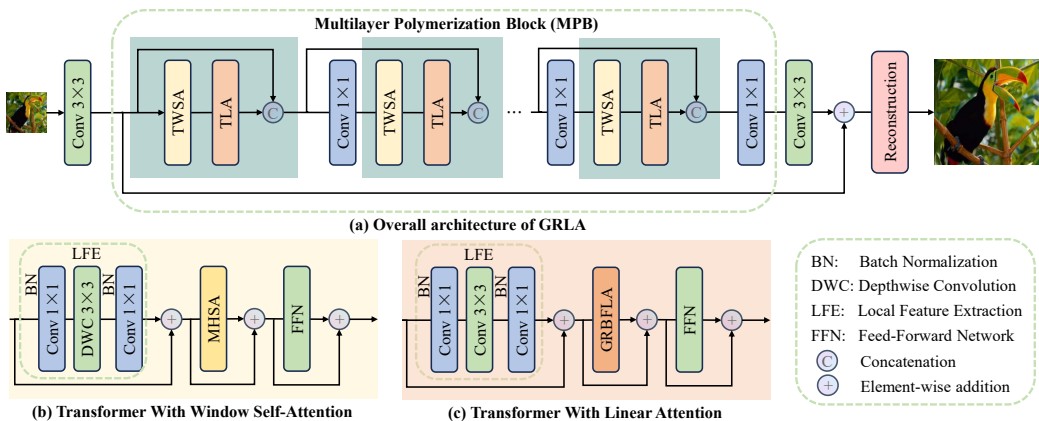

Figure 4: Schematic illustration of the proposed Gaussian Radial Basis Function (GRBF)-based Linear Attention (GRLA). (a) Overall architecture of GRLA. (b) Architecture of the TWSA. (c) Architecture of the TLA.

weighted by the norm-related terms of key vectors, containing spatial structural information, as it captures the correlation between key vectors $K_j$ and value vectors $V_j$ via norm-based weighting. $\sum_{j=1}^{N} \varphi(K_j)$ denotes the normalization denominator sum weighted by the norm-related terms of key vectors, and this sum serves to scale the attention output, ensuring the magnitude of $\alpha_i$ remains within a reasonable range. And $\sum_{j=1}^{N} \varphi(K_j)K_j$ represents the key vector sum weighted by the norm-related terms of key vectors, and this sum is used for normalizing the key-value interaction component, ensuring consistent scaling with the attention output.

In summary, by introducing the standard Gaussian Radial Basis Function (GRBF) kernel and constructing its linearly computable approximate form $K_{\text{GRBF}}(\cdot)$, we successfully propose a novel GRBF-based Linear Attention (GRBFLA) mechanism. The core of this GRBFLA mechanism lies in its adoption of an approximate kernel function, one rooted in the inherently distance-aware standard GRBF kernel. Through rigorous mathematical derivation and first-order Taylor approximation, the GRBFLA mechanism is successfully adapted to the linear attention computational framework, ultimately achieving a linear computational complexity $O(n)$. Both theoretical analysis and experimental validation demonstrate that the GRBFLA kernel can effectively approximate the behavior of standard Softmax attention, thereby achieving an excellent balance between reconstruction performance and computational efficiency in lightweight image SR tasks.

## 3.4 OVERALL NETWORK ARCHITECTURE

The proposed GRBF-based Linear Attention (GRBFLA) mechanism can effectively capture long-range dependencies while maintaining a linear computational complexity. However, relying solely on long-range dependency capture is insufficient for preserving fine-grained image details, as local feature interactions are equally crucial for high-quality image super-resolution (SR) tasks. Therefore, we integrate a window-based Multi-Head Self-Attention (MHSA) mechanism (Liu et al., 2021) to enhance local feature interactions within non-overlapping windows, thereby compensating for the potential deficiency of pure linear attention in capturing local high-frequency details. To further strengthen local feature interactions, we introduce lightweight convolutional layers before the MHSA and GRBFLA modules; these layers serve to enhance the local correlation of input feature maps, laying a better foundation for subsequent attention-based feature processing. Together, the MHSA module, GRBFLA module, and convolutional layers constitute the core basic elements of the GRLA architecture; this architecture can effectively capture both local and long-range dependencies, while learning complex nonlinear mappings from low-resolution (LR) to high-resolution (HR) features.

As illustrated in Fig. 4, given a low-resolution image as input, GRLA first employs a shallow convolutional layer to extract shallow features; these features encapsulate basic local structural in-

Table 1: Ablation on the effectiveness of different linear kernel functions.

| Linear Kernel | Params (K) | FLOPs (G) | Set5 | | Urban100 | | Manga109 | |
| --- | --- | --- | --- | --- | --- | --- | --- | --- |
| | | | PSNR | SSIM | PSNR | SSIM | PSNR | SSIM |
| Efficient Linear Attention | 885 | 56.5 | 32.58 | 0.8996 | 26.83 | 0.8072 | 31.38 | 0.9190 |
| Taylor Linear Attention | 885 | 56.5 | 32.57 | 0.8998 | 26.88 | 0.8079 | 31.43 | 0.9191 |
| ReLU Linear Attention | 885 | 56.5 | 32.62 | 0.8998 | 26.87 | 0.8081 | 31.32 | 0.9188 |
| Magnitude-Aware Linear Attention | 922 | 58.7 | 32.62 | 0.9000 | 26.85 | 0.8086 | 31.41 | 0.9195 |
| **GRBF Linear Attention** | 885 | 56.5 | **32.64** | **0.9001** | **26.94** | **0.8098** | **31.49** | **0.9200** |

Table 2: Ablation on the effectiveness of $\varphi(K_j)$.

| $\varphi(K_j)$ | Params (K) | FLOPs (G) | Set5 | | Set14 | | BSD100 | |
| --- | --- | --- | --- | --- | --- | --- | --- | --- |
| | | | PSNR | SSIM | PSNR | SSIM | PSNR | SSIM |
| ✗ | 885 | 56.5 | – | – | – | – | – | – |
| ✔ | 885 | 56.5 | **32.64** | **0.9001** | **28.89** | **0.7880** | **27.78** | **0.7437** |

formation of the input image. These extracted shallow features are then fed into multiple Multi-layer Polymerization Blocks (MPBs), a core component of the GRLA network responsible for hierarchical feature processing. Each MPB employs a synergistic design of TWSA and TLA to model dependencies from local to long-range, forming an image hierarchy. A feed-forward network (FFN) further transforms and enhances the features, creating richer representations. To fully leverage features from different levels, we introduce multi-layer aggregation connections to fuse features generated by different MPB layers, enhancing feature expressiveness and improving final SR performance. However, multi-layer aggregation connections inevitably increase model size and computational resource consumption. To mitigate this, we use $1 \times 1$ convolutional layers to adaptively fuse aggregated features, obtaining a more compact representation. These layers learn weight relationships between features at different levels, enabling adaptive fusion. Through this network design, our method achieves efficient feature extraction and reconstruction in image SR.

## 4 EXPERIMENTS

### 4.1 ABLATION STUDY

**Impact of Different Linear Kernels**: To evaluate the impact of different linear kernel functions on model complexity, computational overhead, and super-resolution performance, we conduct a systematic comparison of four representative linear attention methods. These methods include Efficient Linear Attention (Shen et al., 2021), Taylor Linear Attention (Qiu et al., 2023), Relu Linear Attention (Cai et al., 2023), and Magnitude-Aware Linear Attention (Fan et al., 2025) , which are widely cited in linear attention research. For fair comparison, we maintain all other experimental settings (e.g., model structure, training parameters, dataset configuration) unchanged. We only replace the core GRBF-based Linear Attention (GRBFLA) module in the GRLA network with the core linear attention module of the four comparison methods. Table 1 presents quantitative comparisons of all methods. Among the four compared methods, MALA has more parameters and higher computational overhead yet still underperforms our GRLA-based method in SR performance. Our GRLA network outperforms all four variants in key metrics (peak signal-to-noise ratio (PSNR), structural similarity index measure (SSIM)), fully validating the effectiveness and necessity of GRLA's inherently distance-aware GRBF kernel.

**Impact of Norm Terms in GRBF Kernel**: To investigate the specific role of norm-related terms in the GRBF kernel (adopted in GRLA), we construct a comparative model by removing the norm-related term $\varphi(K_j)$ from the original GRLA network. Table 2 presents the experimental results of the original GRLA network and the comparative model (without $\varphi(K_j)$ on five standard benchmark datasets for ×4 image super-resolution tasks. The experimental results show that when the norm-related term $\varphi(K_j)$ is removed, the comparison model cannot achieve effective training convergence and experiences gradient explosion. This proves that $\varphi(K_j)$ effectively encodes the L2 norm information of the K vector, thereby effectively suppressing gradient fluctuations and ensuring stable

Table 3: Quantitative comparison on lightweight image super-resolution with state-of-the-art methods. The best and the second best results are in red and blue, respectively.

| Method | Scale | Params (K) | FLOPs (G) | Set5 PSNR | Set5 SSIM | Set14 PSNR | Set14 SSIM | BSD100 PSNR | BSD100 SSIM | Urban100 PSNR | Urban100 SSIM | Manga109 PSNR | Manga109 SSIM |
|---|---|---|---|---|---|---|---|---|---|---|---|---|---|
| Bicubic | | - | - | 33.66 | 0.9299 | 30.24 | 0.8688 | 29.56 | 0.8431 | 26.88 | 0.8403 | 30.80 | 0.9339 |
| IDN (Hui et al., 2018) | | 553 | 124.6 | 37.83 | 0.9600 | 33.30 | 0.9148 | 32.08 | 0.8985 | 31.27 | 0.9196 | 38.01 | 0.9749 |
| CARN (Ahn et al., 2018) | | 1592 | 222.8 | 37.76 | 0.9590 | 33.52 | 0.9166 | 32.09 | 0.8978 | 31.92 | 0.9256 | 38.36 | 0.9765 |
| LAPAR-A (Li et al., 2020) | | 548 | 171.0 | 38.01 | 0.9605 | 33.62 | 0.9183 | 32.19 | 0.8999 | 32.10 | 0.9283 | 38.67 | 0.9772 |
| IMDN (Hui et al., 2019) | | 694 | 158.8 | 38.00 | 0.9605 | 33.63 | 0.9177 | 32.19 | 0.8996 | 32.17 | 0.9283 | 38.88 | 0.9774 |
| RFDN (Liu et al., 2020) | | 534 | 95.0 | 38.05 | 0.9606 | 33.68 | 0.9184 | 32.16 | 0.8994 | 32.12 | 0.9278 | 38.88 | 0.9773 |
| LatticeNet (Luo et al., 2020) | ×2 | 756 | 169.5 | 38.15 | 0.9610 | 33.78 | 0.9193 | 32.25 | 0.9005 | 32.43 | 0.9302 | 38.94 | 0.9773 |
| SwinIR-light (Liang et al., 2021) | | 910 | 244.2 | 38.14 | 0.9611 | 33.86 | 0.9206 | 32.31 | 0.9012 | 32.76 | 0.9340 | 39.12 | 0.9783 |
| ELAN (Zhang et al., 2022) | | 621 | 203.1 | 38.17 | 0.9611 | 33.94 | 0.9207 | 32.30 | 0.9012 | 32.76 | 0.9340 | 39.11 | 0.9782 |
| MambaIR-light (Guo et al., 2024) | | 905 | 334.2 | 38.13 | 0.9610 | 33.95 | 0.9208 | 32.31 | 0.9013 | 32.85 | 0.9349 | 39.20 | 0.9782 |
| SRFormer-light (Zhou et al., 2023) | | 853 | 236.3 | 38.23 | 0.9613 | 33.94 | 0.9209 | 32.36 | 0.9019 | 32.91 | 0.9353 | 39.28 | 0.9785 |
| DCTLSA (Zeng et al., 2023) | | 867 | 203.9 | 38.25 | 0.9612 | 34.03 | 0.9219 | 32.37 | 0.9020 | 32.96 | 0.9362 | 39.33 | 0.9781 |
| ESC-lt (Lee et al., 2025) | | 603 | 359.4 | 38.24 | 0.9615 | 33.98 | 0.9211 | 32.35 | 0.9020 | 33.05 | 0.9363 | 39.33 | 0.9786 |
| MambaIRv2-light (Guo et al., 2025) | | 774 | 286.3 | 38.26 | 0.9615 | 34.09 | 0.9221 | 32.36 | 0.9019 | 33.26 | 0.9378 | 39.35 | 0.9785 |
| **GRLA (Ours)** | | 867 | 213.5 | 38.33 | 0.9616 | 34.14 | 0.9236 | 32.37 | 0.9019 | 33.10 | 0.9367 | 39.48 | 0.9784 |
| Bicubic | | - | - | 30.39 | 0.8682 | 27.55 | 0.7742 | 27.21 | 0.7385 | 24.46 | 0.7349 | 26.95 | 0.8556 |
| IDN (Hui et al., 2018) | | 553 | 56.3 | 34.11 | 0.9253 | 29.99 | 0.8354 | 28.95 | 0.8013 | 27.42 | 0.8359 | 32.71 | 0.9381 |
| CARN (Ahn et al., 2018) | | 1592 | 118.8 | 34.29 | 0.9255 | 30.29 | 0.8407 | 29.06 | 0.8034 | 28.06 | 0.8493 | 33.50 | 0.9440 |
| LAPAR-A (Li et al., 2020) | | 544 | 114.0 | 34.36 | 0.9267 | 30.34 | 0.8421 | 29.11 | 0.8054 | 28.15 | 0.8523 | 33.51 | 0.9441 |
| IMDN (Hui et al., 2019) | | 703 | 71.5 | 34.36 | 0.9270 | 30.32 | 0.8417 | 29.09 | 0.8046 | 28.17 | 0.8519 | 33.61 | 0.9445 |
| RFDN (Liu et al., 2020) | | 541 | 42.2 | 34.41 | 0.9273 | 30.34 | 0.8420 | 29.09 | 0.8050 | 28.21 | 0.8525 | 33.67 | 0.9449 |
| LatticeNet (Luo et al., 2020) | ×3 | 765 | 76.3 | 34.53 | 0.9281 | 30.39 | 0.8424 | 29.15 | 0.8059 | 28.33 | 0.8538 | 33.63 | 0.9441 |
| SwinIR-light (Liang et al., 2021) | | 918 | 111.2 | 34.62 | 0.9289 | 30.54 | 0.8463 | 29.20 | 0.8082 | 28.66 | 0.8624 | 33.98 | 0.9478 |
| ELAN (Zhang et al., 2022) | | 629 | 90.1 | 34.61 | 0.9288 | 30.55 | 0.8463 | 29.21 | 0.8081 | 28.69 | 0.8624 | 34.00 | 0.9478 |
| MambaIR-light (Guo et al., 2024) | | 913 | 148.5 | 34.63 | 0.9288 | 30.54 | 0.8459 | 29.23 | 0.8084 | 28.70 | 0.8631 | 34.12 | 0.9479 |
| SRFormer-light (Zhou et al., 2023) | | 861 | 105.4 | 34.67 | 0.9296 | 30.57 | 0.8469 | 29.26 | 0.8099 | 28.81 | 0.8655 | 34.19 | 0.9489 |
| DCTLSA (Zeng et al., 2023) | | 874 | 90.6 | 34.70 | 0.9292 | 30.59 | 0.8466 | 29.26 | 0.8091 | 28.78 | 0.8650 | 34.34 | 0.9489 |
| ESC-lt (Lee et al., 2025) | | 612 | 162.8 | 34.61 | 0.9295 | 30.52 | 0.8475 | 29.26 | 0.8102 | 28.93 | 0.8679 | 34.33 | 0.9495 |
| MambaIRv2-light (Guo et al., 2025) | | 781 | 126.7 | 34.71 | 0.9298 | 30.68 | 0.8483 | 29.26 | 0.8098 | 29.01 | 0.8689 | 34.41 | 0.9497 |
| **GRLA (Ours)** | | 874 | 94.9 | 34.80 | 0.9304 | 30.70 | 0.8483 | 29.31 | 0.8107 | 29.07 | 0.8695 | 34.53 | 0.9504 |
| Bicubic | | - | - | 28.42 | 0.8104 | 26.00 | 0.7027 | 25.96 | 0.6675 | 23.14 | 0.6577 | 24.89 | 0.7866 |
| IDN (Hui et al., 2018) | | 553 | 32.3 | 31.82 | 0.8903 | 28.25 | 0.7730 | 27.41 | 0.7297 | 25.41 | 0.7632 | 29.41 | 0.8942 |
| CARN (Ahn et al., 2018) | | 1592 | 90.9 | 32.13 | 0.8937 | 28.60 | 0.7806 | 27.58 | 0.7349 | 26.07 | 0.7837 | 30.47 | 0.9084 |
| LAPAR-A (Li et al., 2020) | | 659 | 94.0 | 32.15 | 0.8944 | 28.61 | 0.7818 | 27.61 | 0.7366 | 26.14 | 0.7871 | 30.42 | 0.9074 |
| IMDN (Hui et al., 2019) | | 715 | 40.9 | 32.21 | 0.8948 | 28.58 | 0.7811 | 27.56 | 0.7353 | 26.04 | 0.7838 | 30.45 | 0.9075 |
| RFDN (Liu et al., 2020) | | 550 | 23.9 | 32.24 | 0.8952 | 28.61 | 0.7819 | 27.57 | 0.7360 | 26.11 | 0.7858 | 30.58 | 0.9089 |
| LatticeNet (Luo et al., 2020) | ×4 | 777 | 43.6 | 32.30 | 0.8962 | 28.68 | 0.7830 | 27.62 | 0.7367 | 26.25 | 0.7873 | 30.54 | 0.9073 |
| SwinIR-light (Liang et al., 2021) | | 930 | 63.6 | 32.44 | 0.8976 | 28.77 | 0.7858 | 27.69 | 0.7406 | 26.47 | 0.7980 | 30.92 | 0.9151 |
| ELAN (Zhang et al., 2022) | | 640 | 54.1 | 32.43 | 0.8975 | 28.78 | 0.7858 | 27.69 | 0.7406 | 26.54 | 0.7982 | 30.92 | 0.9150 |
| MambaIR-light (Guo et al., 2024) | | 924 | 84.6 | 32.42 | 0.8977 | 28.74 | 0.7847 | 27.68 | 0.7400 | 26.52 | 0.7983 | 30.94 | 0.9135 |
| SRFormer-light (Zhou et al., 2023) | | 873 | 62.8 | 32.51 | 0.8988 | 28.82 | 0.7872 | 27.73 | 0.7422 | 26.67 | 0.8032 | 31.17 | 0.9165 |
| DCTLSA (Zeng et al., 2023) | | 885 | 53.9 | 32.52 | 0.8987 | 28.82 | 0.7869 | 27.73 | 0.7421 | 26.70 | 0.8045 | 31.14 | 0.9165 |
| ESC-lt (Lee et al., 2025) | | 624 | 91.0 | 32.52 | 0.8995 | 28.87 | 0.7878 | 27.72 | 0.7423 | 26.76 | 0.8058 | 31.26 | 0.9173 |
| MambaIRv2-light (Guo et al., 2025) | | 790 | 75.6 | 32.51 | 0.8992 | 28.84 | 0.7878 | 27.75 | 0.7426 | 26.82 | 0.8079 | 31.24 | 0.9182 |
| **GRLA (Ours)** | | 885 | 56.5 | 32.64 | 0.9001 | 28.89 | 0.7880 | 27.78 | 0.7437 | 26.94 | 0.8098 | 31.49 | 0.9200 |

convergence of the model. This phenomenon fully demonstrates the key role of $\varphi(K_j)$ in maintaining the numerical stability of the GRLA network and preserving the distance-aware information of the GRBF kernel.

## 4.2 COMPARATIVE EVALUATION

We conduct a comprehensive comparison between the proposed GRLA (the implementation details can be found in Appendix A.2) model and eleven representative SR methods, including the classic bicubic interpolation baseline and ten advanced lightweight architectures: IDN (Hui et al., 2018), CARN (Ahn et al., 2018), LAPAR-A (Li et al., 2020), IMDN (Hui et al., 2019), RFDN (Liu et al., 2020), LatticeNet (Luo et al., 2020), SwinIR-light (Liang et al., 2021), ELAN (Zhang et al., 2022), SRFormer-light (Zhou et al., 2023), DCTLSA (Zeng et al., 2023), ESC-lt (Lee et al., 2025), MambaIR-light (Guo et al., 2024), and MambaIRv2-light (Guo et al., 2025). All experiments follow the reproducibility protocol for reliability and fairness. See Appendix A.1 for dataset and evaluation metrics details. The selected comparison methods have demonstrated excellent performance in previous studies, providing a competitive benchmark. We analyze GRLA's effectiveness from quantitative results, visual quality, and model efficiency.

**Quantitative Comparison**: Table 3 reports objective quantitative metrics on five benchmark datasets. Results show that Transformer-based architectures generally outperform traditional CNN methods, benefiting from the self-attention mechanism's advantage in modeling long-range dependencies. Our proposed GRLA achieves the best performance on almost all datasets and upsampling factors (×2, ×3, ×4). Specifically, on Manga109, GRLA improves PSNR by 0.13 dB, 0.12 dB, and 0.25 dB for ×2, ×3, and ×4 SR tasks, respectively, compared to the second-best MambaIRv2-light. This significant improvement validates the effectiveness of GRLA's core module design: the introduced GRBFLA module effectively models long-range dependencies. Moreover, from the com-

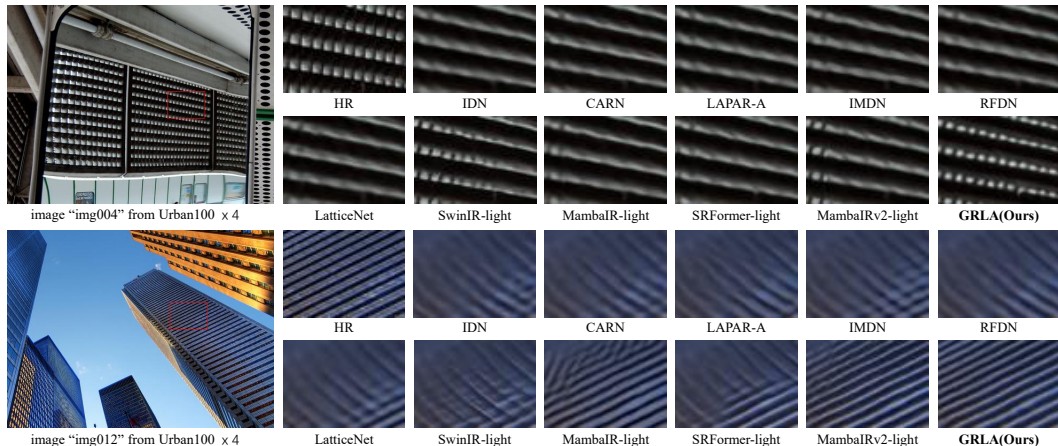

Figure 5: Qualitative comparison of our GRLA with different methods on Urban100 ×4 lightweight image SR.

parison of model parameters and computational cost in Table 3, GRLA achieves performance break-throughs under lightweight constraints with fewer parameters than SwinIR-light and lower computation than MambaIRv2-light, demonstrating excellent balance between efficiency and performance. The core innovation of GRLA lies in its GRBF-based linear attention module (GRBFLA), which approximates the long-range dependency modeling capability of standard Softmax attention while maintaining linear complexity, avoiding quadratic computational costs. Thus, GRLA achieves better reconstruction accuracy with comparable parameters and computational cost. In summary, these experiments systematically verify that GRLA achieves state-of-the-art performance in lightweight super-resolution tasks.

**Qualitative Comparison**: We qualitatively compare the proposed GRLA with current mainstream lightweight SR methods. Visual comparison results show that GRLA exhibits significant advantages in reconstructing image details, with higher fidelity in high-frequency textures than all compared models. Specifically, as illustrated in Fig. 5, on images "img004" and "img012" from Urban100, GRLA more accurately reconstructs edges and contours, while other models generally exhibit blurry edges, structural distortion, or fragmentation. These visual comparisons demonstrate GRLA's clear advantage in recovering high-frequency structures and detail information from low-resolution inputs. The quantitative analysis and visual results jointly validate the effectiveness of GRLA, showing that it achieves excellent reconstruction quality while maintaining low computational complexity. This research provides a practical and efficient solution for developing lightweight super-resolution models suitable for real-world scenarios. For more evaluations, please refer to Appendix A.4.

## 5 CONCLUSION

This paper proposes a lightweight image super-resolution (SR) framework called Gaussian Radial Basis Function (GRBF)-based Linear Attention (GRLA). Its core innovation is the introduction of a distance-aware GRBF kernel, which underpins the framework's attention computation. Mathematical derivation verifies the equivalence between the proposed GRBF kernel and standard Softmax attention, justifying its use as a substitute in lightweight SR tasks. Via first-order Taylor approximation, the GRBF kernel is transformed into a linearly computable, distance-aware form ($K_{\text{GRBF}(\cdot)}$) that approximates Softmax attention. Based on this kernel, we construct the efficient GRBF-based Linear Attention (GRBFLA) module, GRLA's core component, which enables linear complexity long-range dependency modeling. Experiments on multiple benchmarks show GRLA outperforms existing lightweight SR methods in both reconstruction quality and computational efficiency. Future work will explore its generalization in downstream tasks (e.g., video SR, object detection) and optimize the model structure for better performance.

**Reproducibility Statement**: The models, environments, core parameters, and dataset processing methods used in the experiments of this study have all been clearly documented to ensure the reproducibility of the results.

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

# A APPENDIX

## A.1 DATASETS AND EVALUATION METRICS

We train our models using the widely adopted DIV2K (Timofte et al., 2017) dataset, which contains 800 pairs of high-resolution (HR) and low-resolution (LR) images. To comprehensively evaluate the performance of the proposed GRLA method, we conduct systematic experiments on five standard test sets: Set5 (Bevilacqua et al., 2012), Set14 (Zeyde et al., 2010), BSD100 (Martin et al., 2001), Urban100 (Huang et al., 2015), and Manga109 (Matsui et al., 2017). Evaluation strictly follows common practices in the field: all results are computed on the luminance channel (Y channel) in YCbCr color space, using peak signal-to-noise ratio (PSNR) and structural similarity index measure (SSIM) as quantitative metrics.

## A.2 IMPLEMENTATION DETAILS

During training, we adopt a patch-based random sampling strategy: each LR input image is randomly cropped into 16 patches of size 64×64. This strategy ensures training efficiency while preserving local context information. To further improve generalization, we use data augmentation including rotations (90°, 180°, 270°) and horizontal flipping. The GRLA network uses a lightweight architecture with the number of channels set to 55 and the number of MPB modules set to 6 (specific hyperparameters are determined via cross-validation; see ablation study on the impact of MPB count). Optimization uses the Adam optimizer with hyperparameters $\beta_1 = 0.9$, $\beta_2 = 0.999$, $\epsilon = $ 1e-8, trained for 1000 epochs. The initial learning rate is 5e-4, halved every 200 epochs. These settings effectively balance training stability and final model performance.

Table 4: Ablation on the effectiveness of $\gamma$ value.

| $\gamma(\times\sqrt{d})$ | Params (K) | FLOPs (G) | Urban100 | | Manga109 | |
|---|---|---|---|---|---|---|
| | | | PSNR | SSIM | PSNR | SSIM |
| 1 | 885 | 56.5 | – | – | – | – |
| 2/3 | 885 | 56.5 | – | – | – | – |
| 1/2 | 885 | 56.5 | **26.94** | **0.8098** | **31.49** | **0.9200** |
| 1/4 | 885 | 56.5 | 26.88 | 0.8078 | 31.44 | 0.9191 |
| 1/8 | 885 | 56.5 | 26.87 | 0.8089 | 31.49 | 0.9200 |
| 1/16 | 885 | 56.5 | 26.88 | 0.8080 | 31.37 | 0.9190 |

Table 5: Higher-order Taylor Approximation.

| Taylor approximation | Memory (MB) | Set14 | | Urban100 | | Manga109 | |
|---|---|---|---|---|---|---|---|
| | | PSNR | SSIM | PSNR | SSIM | PSNR | SSIM |
| Second-order | 13266 | 28.87 | 0.7879 | 26.90 | 0.8086 | 31.48 | 0.9199 |
| **First-order** | **12144** | **28.89** | **0.7880** | **26.94** | **0.8098** | **31.49** | **0.9200** |

### A.3 Additional Ablation Studies

**Impact of $\gamma$ value**: To systematically evaluate the impact of the bandwidth parameter $\gamma$ on SR reconstruction quality, we design six different $\gamma$ configurations. $\gamma$ is a key hyperparameter of the GRBF kernel that controls the influence of Euclidean distance on the kernel's similarity calculation. As presented in Table 4, our GRLA-based model achieves optimal SR reconstruction performance when $\gamma$ is set to 1/2. This specific $\gamma$ value (1/2) not only satisfies the first-order Taylor approximation condition (introduced in Section 3.2) but also maximizes the distance-aware capability of the GRBF kernel. However, when $\gamma$ increases to 2/3 or 1, the model's loss function exhibits drastic oscillations. These oscillations hinder the model from achieving stable training convergence. Based on the above experimental results, we set the default value of the bandwidth parameter $\gamma$ to 1/2. This default setting enables the GRLA model to achieve optimal SR reconstruction performance while maintaining stable training processes.

**Taylor Approximation Error Analysis:** Through normalization and the setting of bandwidth parameter $\gamma$, the maximum value of $2\gamma Q_i^T K_j$ is $2 \times \left(\frac{1}{2\sqrt{d}}\right) \times 1 = \frac{1}{\sqrt{d}}$. Under the configuration $d = 55$ in this study, this value is approximately 0.134. As illustrated in Figure 6(a), we statistically analyzed the distribution of $2\gamma Q_i^T K_j$ for random input data. Over 98% of the samples fall within the interval $[-0.2, 0.2]$, where the relative error of the first-order Taylor approximation is $\leq 2\%$ (calculated as $\frac{|e^x-(1+x)|}{e^x}$; the error is approximately 2% when $x = 0.2$). This demonstrates the effectiveness of the approximation under practical input distributions. For $x = 2\gamma Q_i^T K_j$, the Lagrange remainder of the first-order Taylor approximation is $R_1(x) = \frac{e^\xi x^2}{2}$ (where $\xi \in [0, x]$). Incorporating the maximum value of $x = 0.134$, we obtain $R_1(x) \leq \frac{e^{0.134} \times (0.134)^2}{2} \approx \frac{1.143 \times 0.018}{2} \approx 0.0103$, indicating an absolute error $\leq 0.0103$ and a relative error $\leq \frac{0.0103}{e^{0.134}} \approx 0.009$. This confirms that the approximation error is at an extremely low level. Meanwhile, as shown in Figure 6(b), which presents the error distribution across different input samples, the approximation error has a mean of 0.0042 and a standard deviation of 0.0052, further verifying the stability and controllability of the error.

**Higher-order Taylor Approximation:** To investigate the influence of higher-order Taylor approximations, we studied the effects of different-order Taylor expansions. We performed first-order and second-order Taylor expansions on GRBFLA. Table 5 shows that the first-order Taylor expansion achieved better results than the second-order Taylor expansion, and its memory usage was significantly lower than that of the second-order Taylor expansion. Therefore, we ultimately chose the first-order Taylor expansion.

**Impact of TLA**: TLA is the core module of GRLA, playing a key role in long-range dependency modeling based on linear attention. To evaluate the contribution of different submodules, we conduct

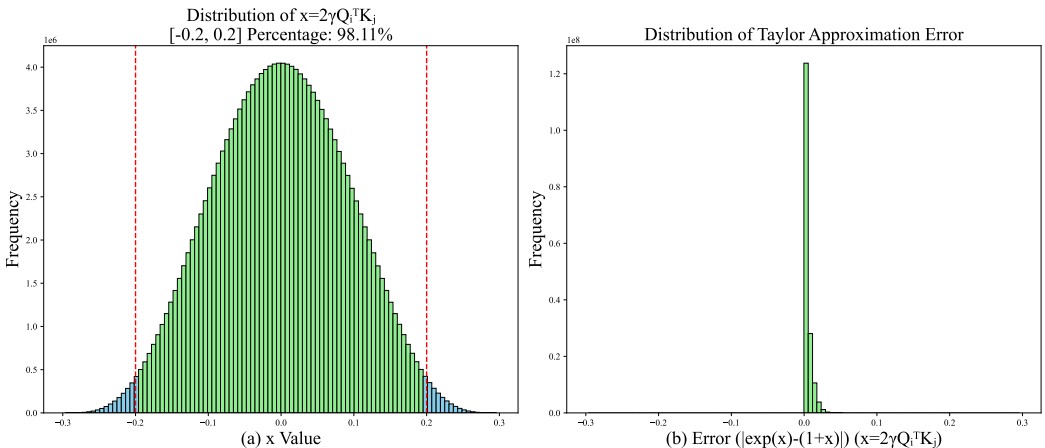

Figure 6: Taylor error distribution.

Table 6: Ablation on the effectiveness of Transformer With Linear Attention (TLA).

| TMHSA | TLA | Set14 | | B100 | | Manga109 | |
|---|---|---|---|---|---|---|---|
| | | PSNR | SSIM | PSNR | SSIM | PSNR | SSIM |
| ✔(8) | ✗ | 28.83 | 0.7864 | 27.70 | 0.7404 | 30.94 | 0.9143 |
| ✔(8) | ✔ | 28.87 | 0.7873 | 27.76 | 0.7422 | 31.32 | 0.9184 |
| ✔(16) | ✗ | 28.79 | 0.7858 | 27.69 | 0.7406 | 31.06 | 0.9156 |
| ✔(16) | ✔ | **28.89** | **0.7880** | **27.78** | **0.7437** | **31.49** | **0.9200** |

ablation experiments with three configurations: (1) remove TLA, reverting to standard self-attention (window sizes 8 and 16); (2) use our full proposed scheme. As shown in Table 6, using only window attention limits the receptive field to local windows, restricting performance. Introducing TLA with distance-aware global modeling capability brings a significant PSNR improvement of 0.43 dB on Manga109, verifying its effectiveness and necessity.

**Impact of Channel Number**: We conduct ×4 SR experiments on Set5 and Manga109 to study the impact of channel number on reconstructed image quality. Quantitative results in Fig. 7 show that network parameters and computational cost increase monotonically with channel number. PSNR peaks at 55 channels and then gradually decreases. To keep model complexity comparable to mainstream methods (e.g., SwinIR-light (Liang et al., 2021), MambaIR-light (Guo et al., 2024), MambaIRv2-light (Guo et al., 2025)) and balance performance and efficiency, we set the default channel number to 55.

**Impact of MPB Number**: To investigate the impact of the number of Multi-layer Polymerization Blocks (MPBs) on SR performance, we conduct comparative experiments with 4, 5, 6, 7 MPB modules under ×4 SR. As illustrated in Fig. 8, results on BSD100 and Urban100 show that model parameters and FLOPs increase monotonically with the number of MPBs. Notably, when the number of MPBs is set to 6, the performance growth rate slows down, while the model size remains comparable to lightweight methods such as SwinIR-light and MambaIRv2-light, achieving the optimal performance. Based on a trade-off between performance and complexity, we set the default number of MPBs to 6.

**Impact of Multi-layer Aggregation Connections**: To explore the effectiveness of multi-layer aggregation connections, we build a comparative model without any 1×1 convolutional layers (keeping MPB count at 6). Table 7 reports quantitative results for ×4 SR. Experiments show that introducing multi-layer aggregation connections significantly improves model performance, verifying that multi-layer connections in GRLA effectively integrate multi-level information and enhance salient feature extraction. However, these connections also increase parameters and computational cost. Thus, GRLA's SR model must balance computational efficiency and performance gains. After opti-

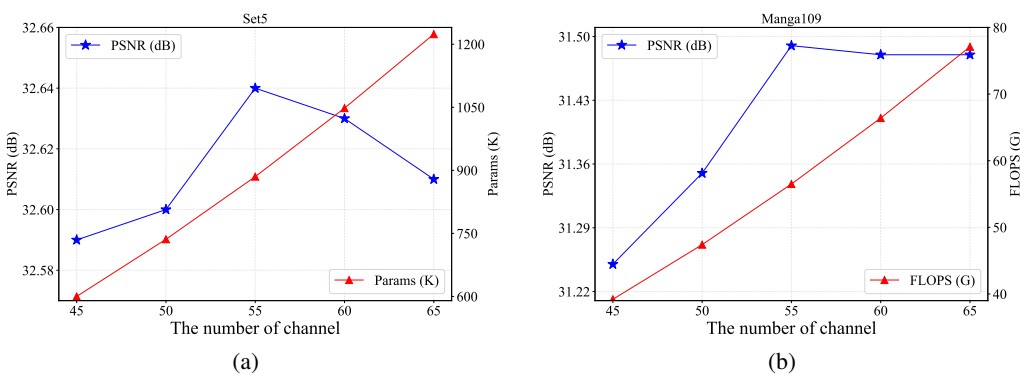

Figure 7: Ablation on the effectiveness of channel number.

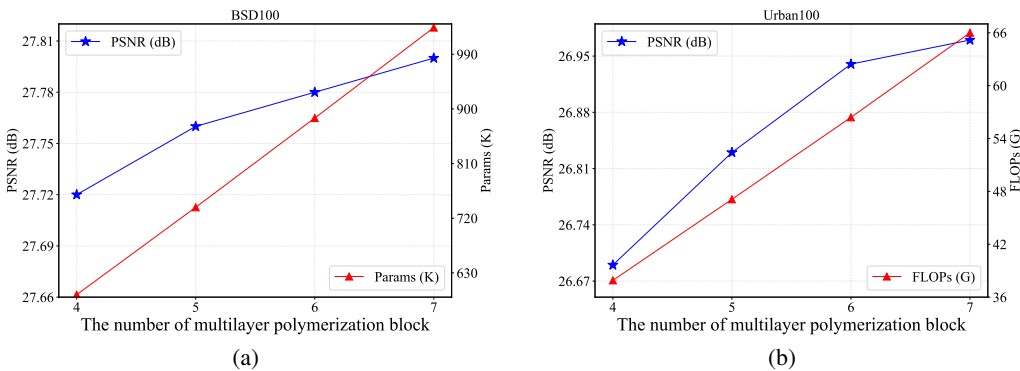

Figure 8: Ablation on the effectiveness of Multilayer Polymerization Block (MPB) number.

Table 7: Ablation on the effectiveness of multi-layer aggregation connections.

| Multi-layer Aggregation | Params (K) | FLOPs (G) | Set5 | | Set14 | | Manga109 | |
|---|---|---|---|---|---|---|---|---|
| | | | PSNR | SSIM | PSNR | SSIM | PSNR | SSIM |
| ✗ | 824 | 53.0 | 32.63 | 0.8996 | 28.85 | 0.7873 | 28.85 | 0.7873 |
| ✔ | 885 | 56.5 | **32.64** | **0.9001** | **28.89** | **0.7880** | **28.89** | **0.7880** |

Table 8: Quantitative comparison on lightweight image super-resolution with other methods.

| Method | Scale | Params (K) | FLOPs (G) | Set5 | | Set14 | | BSD100 | | Urban100 | | Manga109 | |
|---|---|---|---|---|---|---|---|---|---|---|---|---|---|
| | | | | PSNR | SSIM | PSNR | SSIM | PSNR | SSIM | PSNR | SSIM | PSNR | SSIM |
| Bicubic | ×2 | - | - | 33.66 | 0.9299 | 30.24 | 0.8688 | 29.56 | 0.8431 | 26.88 | 0.8403 | 30.80 | 0.9339 |
| IDN | ×2 | 553 | 124.6 | 37.83 | 0.9600 | 33.30 | 0.9148 | 32.08 | 0.8985 | 31.27 | 0.9196 | 38.01 | 0.9749 |
| CARN | ×2 | 1592 | 222.8 | 37.76 | 0.9590 | 33.52 | 0.9166 | 32.09 | 0.8978 | 31.92 | 0.9256 | 38.36 | 0.9765 |
| LAPAR-A | ×2 | 548 | 171.0 | 38.01 | 0.9605 | 33.62 | 0.9183 | 32.19 | 0.8999 | 32.10 | 0.9283 | 38.67 | 0.9772 |
| IMDN | ×2 | 694 | 158.8 | 38.00 | 0.9605 | 33.63 | 0.9177 | 32.19 | 0.8996 | 32.17 | 0.9283 | 38.88 | 0.9774 |
| RFDN | ×2 | 534 | 95.0 | 38.05 | 0.9606 | 33.68 | 0.9184 | 32.16 | 0.8994 | 32.12 | 0.9278 | 38.88 | 0.9773 |
| LatticeNet | ×2 | 756 | 169.5 | 38.15 | 0.9610 | 33.78 | 0.9193 | 32.25 | 0.9005 | 32.43 | 0.9302 | 38.94 | 0.9773 |
| SwinIR-light | ×2 | 910 | 244.2 | 38.14 | 0.9611 | 33.86 | 0.9206 | 32.31 | 0.9012 | 32.76 | 0.9340 | 39.12 | 0.9783 |
| **Full GRBFLA** | ×2 | 867 | 199.2 | **38.19** | **0.9610** | **33.84** | **0.9201** | **32.27** | **0.9006** | **32.58** | **0.9317** | **39.24** | **0.9774** |

mization, GRLA achieves better performance with fewer parameters and lower computational complexity.

Table 9: Quantitative comparison on classic image super-resolution with SwinIR.

| Method | Scale | Params (M) | Set5 | | Set14 | | BSD100 | | Urban100 | | Manga109 | |
|---|---|---|---|---|---|---|---|---|---|---|---|---|
| | | | PSNR | SSIM | PSNR | SSIM | PSNR | SSIM | PSNR | SSIM | PSNR | SSIM |
| SwinIR | ×4 | 11.9 | 32.72 | 0.9021 | 28.94 | 0.7914 | 27.83 | 0.7459 | 27.07 | 0.8164 | 31.67 | 0.9226 |
| **GRLA-L (Ours)** | ×4 | 7.6 | **32.78** | 0.9019 | **29.00** | 0.7906 | **27.85** | 0.7460 | **27.20** | 0.8175 | **31.72** | 0.9224 |

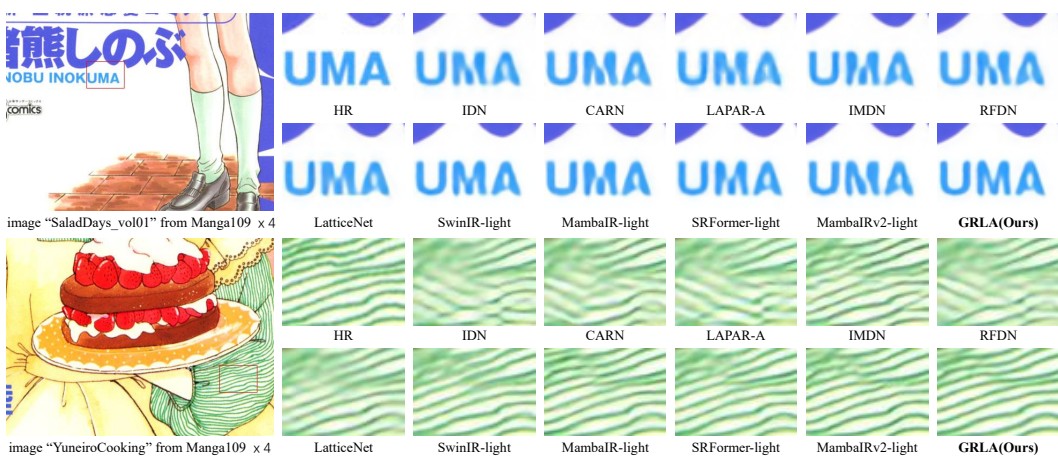

Figure 9: Qualitative comparison of our GRLA with different methods on Manga109 ×4 lightweight image SR.

Table 10: The average inference time on Urban100 dataset.

| Model | SwinIR-light | MambaIR-light | MambaIRv2-light | GRLA |
|---|---|---|---|---|
| Latency (ms) | 213.4 | 208.9 | 388.0 | **60.9** |

## A.4 ADDITIONAL COMPARATIVE EVALUATION

**Full GRBFLA**: We removed TWSA from GRLA and completely used TLA to construct the network, as shown in Table 8. This proves that using only GRBFLA can completely replace the local window attention mechanism.

**Classic Image Super-Resolution**: We conducted tests on the classic super-resolution method to evaluate its generalization ability. To reduce the training time, we constructed a 7M GRLA network, as shown in Table 9, and its performance was superior to that of SwinIR.

**Qualitative Comparison**: Extensive experiments conducted on benchmark datasets (i.e., Set5, Set14, B100, Urban100, and Manga109) indicate that GRLA outperforms existing lightweight SR models in terms of both PSNR/SSIM metrics and computational efficiency.As shown in Table 6, with fewer parameters and computational costs, GRLA achieves a PSNR improvement of up to 0.57 dB compared to SwinIR-light (Liang et al., 2021). Compared to Mamba-based methods (i.e., MambaIR-light (Guo et al., 2024) and MambaIRv2-light (Guo et al., 2025)), GRLA achieves superior reconstruction quality with significantly lower FLOPs, reducing the latter by 33% and 25%, respectively. As illustrated in Fig. 9, on images "SaladDays_vol01" and "YuneiroCooking" from Manga109, GRLA better preserves sharp edges and detailed forms of character strokes and clothing textures. In contrast, other models often show artifacts or distortions, failing to promote the reconstruction of sharp edges and natural textures.

**Latency Comparison**: To evaluate model efficiency, we report the inference latency of GRLA and other methods measured on a workstation with a single NVIDIA GeForce RTX 2080 Ti GPU. Table 10 shows the average runtime on the Urban100 dataset for ×4 scaling. Thanks to the distance-aware linear long-range dependency modeling of GRBFLA, GRLA's inference speed is about 3.5 times faster than SwinIR-light (Liang et al., 2021) and MambaIR-light (Guo et al., 2024), and about 6 times faster than MambaIRv2-light (Guo et al., 2025), enabling real-time inference.

Table 11: Training memory footprint, iteration time, and performance comparison.

| Model | Memory (MB) | Time for 1000 iters (s) | FLOPs (G) | Urban100 | | Manga109 | |
|---|---|---|---|---|---|---|---|
| | | | | PSNR | SSIM | PSNR | SSIM |
| 8 | 8330 | 280 | 53.1 | 26.62 | 0.8019 | 31.26 | 0.9179 |
| 10 | 11342 | 385 | 52.8 | 26.69 | 0.8050 | 31.23 | 0.9171 |
| 12 | 13720 | 449 | 54.3 | 26.90 | 0.8094 | 31.30 | 0.9189 |
| 14 | 15358 | 505 | 55.6 | 26.87 | 0.8091 | 31.40 | 0.9195 |
| 16 | 14690 | 456 | 59.3 | 26.89 | 0.8089 | 31.36 | 0.9191 |
| **GRLA** | **12144** | **375** | **56.5** | **26.94** | **0.8098** | **31.49** | **0.9200** |

**Training Memory Footprint, Iteration Time, and Performance Comparison**: To further evaluate model efficiency, we test the memory footprint and training iteration time of different methods on a workstation with an NVIDIA GeForce RTX 2080 Ti GPU. Five comparative models are constructed by modifying GRLA: replacing its core GRBFLA module with window-based multi-head self-attention (MHSA) only, using window sizes 8, 10, 12, 14, 16. Table 11 reports metrics for ×4 SR on three datasets. Results show that when the window size is $\geq 10$, the comparison models have higher training iteration time than GRLA; when the window size is $\geq 12$, their memory footprint also exceeds GRLA. Benefiting from the efficient design of the GRBFLA module, GRLA significantly reduces memory usage and training iteration time while maintaining excellent reconstruction performance, demonstrating strong potential for lightweight applications.

## A.5 LARGE LANGUAGE MODEL USAGE STATEMENT

This paper has utilized large language models for translation and polishing. The relevant content has undergone manual verification to ensure the accuracy of the core meaning.

