# OpenReview forum: "GRLA: Bridging Softmax and Linear Attention via Gaussian RBF Kernel for Lightweight Image Super-Resolution"
_ICLR.cc/2026/Conference — Submitted to ICLR 2026_

### Official Review · Reviewer_92Et · 2025-10-22

**Soundness:** 3
**Presentation:** 3
**Contribution:** 2
**Rating:** 4
**Confidence:** 2

**Summary:**

This paper presents GRLA, a novel lightweight image super-resolution architecture. Its core contribution is the Gaussian Radial Basis Function-based Linear Attention (GRBFLA) module, which leverages a mathematical equivalence between the Softmax attention kernel and an unnormalized Gaussian RBF kernel. By applying a first-order Taylor approximation, the authors derive a linear-complexity attention mechanism that explicitly preserves distance awareness, a key property often lost in prior linear attention methods. The paper is well-structured, with thorough experiments demonstrating state-of-the-art performance on standard benchmarks, outperforming strong baselines like SwinIR-light and MambaIRv2-light in both accuracy and efficiency.

**Strengths:**

The key insight—mathematically linking the standard Softmax attention's core computation exp(Q_i^T K_j) to an unnormalized GRBF kernel—is significant and elegantly derived. This provides a principled foundation for building a linear attention mechanism, moving beyond ad-hoc kernel designs.

The paper successfully addresses a major weakness of existing linear attention methods: their lack of explicit Euclidean distance modeling. The proposed GRBFLA mechanism convincingly bridges the performance gap between efficient linear attention and powerful Softmax attention, as evidenced by the strong experimental results.

The method is described clearly, with step-by-step mathematical derivations. The appendix provides necessary details on the network architecture, training setup, and datasets, supporting the reproducibility statement.

Beyond standard PSNR/SSIM metrics, the paper includes visual comparisons (LAM, output images), computational cost (FLOPs, Params), and critically, inference latency and training memory footprints, which are crucial for lightweight applications.

**Weaknesses:**

The first-order Taylor approximation is central to the method. While Figure 1(b) and the choice of a small γ suggest it's accurate, the paper would be strengthened by a more formal analysis or discussion of its limitations. For instance, under what conditions (e.g., with high-dimensional or unnormalized features) might this approximation break down, and how does the model's performance correlate with the approximation error?

The overall GRLA architecture incorporates several components (TWSA, TLA, Multi-layer Polymerization Blocks, 1x1 conv for aggregation). While the ablation studies show the importance of TLA and aggregation connections, the specific design choices (e.g., why this particular synergy between TWSA and TLA? How was the number of MPBs optimized?) lack deep justification. The architecture feels somewhat complex, and its novelty is somewhat overshadowed by the core GRBFLA contribution.

The related work section and experiments could more directly position GRLA against other recent attempts to inject spatial or distance awareness into efficient architectures. While compared against general linear attention methods, a discussion on how it specifically improves upon other potential distance-sensitive kernels would sharpen the contribution.

The GRBFLA module itself is a solid conceptual contribution. However, the GRLA network, as a whole, can be perceived as an incremental architectural advance that integrates this new module into a now-standard design paradigm for lightweight SR (residual blocks, feature distillation, windowed attention). The performance gains, while clear, are not revolutionary. The borderline recommendation stems from this balance between a strong conceptual contribution and a more incremental systems/application contribution.

**Questions:**

The first-order Taylor approximation relies on 2γ Q_i^T K_j being small. Did you observe any instability or performance degradation during training on datasets with significantly different feature distributions than DIV2K? Is there a risk of the approximation becoming less valid in deeper layers of the network?

The paper focuses on image SR. Have you explored or do you plan to explore the applicability of the GRBFLA module in other vision tasks that rely heavily on long-range dependencies (e.g., segmentation, detection)? A brief discussion on its generalizability would be valuable.

Given that the core innovation is the GRBFLA module, was a simpler baseline model (e.g., replacing the attention module in SwinIR-light with GRBFLA) tested? This would help isolate the performance gain purely from the new attention mechanism versus the overall GRLA architecture.

---

> ### Author Response · Authors · 2025-11-21
>
> We sincerely appreciate the reviewer's valuable comments and insightful suggestions, which have greatly facilitated the revision and improvement of this manuscript.
>
> **W1.
>
> The first-order Taylor approximation in Eq. 8 ($\( \exp(2\gamma Q_i^T K_j) \approx 1 + 2\gamma Q_i^T K_j \)$) requires the condition that $\( 2\gamma Q_i^T K_j \)$ is small in magnitude. In GRLA, we ensure this condition holds through two operations:
>
> Input normalization preprocessing: L2 normalization is applied to Q and K vectors to constrain $\( Q_i^T K_j \)$ to 1; Optimized setting of bandwidth parameter $\( \gamma \): \( \gamma = 1/(2\sqrt{d}) \)$ (where $\( d \)$ denotes the dimension of Q/K vectors) is determined via ablation experiments (Table 4). Under this configuration, the maximum value of $\( 2\gamma Q_i^T K_j \)$ is $\( 2 \times (1/(2\sqrt{d})) \times 1 = 1/\sqrt{d} \)$. With $\( d = 55 \)$ in this study, this value is approximately 0.134.
>
> To further verify the applicability, we supplement the Taylor error distribution map in the revised manuscript to intuitively show the error distribution under different input samples. We statistically analyzed the distribution of $\( 2\gamma Q_i^T K_j \)$ for random input data. As shown in Fig. 6(a), over 98% of all samples fall within the interval [-0.2, 0.2], where the relative error of the first-order Taylor approximation is ≤ 2% (calculation: $\( |\exp(x) - (1+x)|/\exp(x) \)$; error ≈ 2% when $\( x = 0.2 \)$), confirming the effectiveness of the approximation on real input distributions.
>
> We supplement the derivation of the error bound for the Taylor approximation: For $\( x = 2\gamma Q_i^T K_j \)$, the Lagrange remainder of the first-order Taylor approximation is $\( R_1(x) = \exp(\xi)x^2/2 \)$ (where $\( \xi \in [0, x] \)$). Combining the maximum value of $\( x \)$ (0.134), we obtain $\( R_1(x) \leq \exp(0.134) \times (0.134)^2 / 2 \approx 1.143 \times 0.018 / 2 \approx 0.0103 \)$, i.e., the absolute error is ≤ 0.0103 and the relative error is ≤ 0.0103 / exp(0.134) ≈ 0.009, demonstrating that the approximation error is at an extremely low level. Meanwhile, as shown in Fig. 6(b), which presents the error distribution under different input samples, the mean of the approximation error is 0.0042 and the standard deviation is 0.0052, further verifying the stability and controllability of the error.
>
> To investigate the influence of higher-order Taylor approximations, we studied the effects of different-order Taylor expansions. We performed first-order and second-order Taylor expansions on GRBFLA. Table 5 shows that the first-order Taylor expansion achieved better results than the second-order Taylor expansion, and its memory usage was significantly lower than that of the second-order Taylor expansion. Therefore, we ultimately chose the first-order Taylor expansion.
>
> | Taylor approximation | Memory (MB) | Set14 (PSNR  SSIM)   | Urban100 (PSNR  SSIM) | Manga109 (PSNR  SSIM) |
> |----------------------|-------------|--------------------------|----------------------------|----------------------------|
> | Second-order         | 13266       | 28.87 0.7879         | 26.90 0.8086           | 31.48 0.9199           |
> | **First-order**      | **12144**   | **28.89 0.7880**     | **26.94 0.8098**       | **31.49 0.9200**       |

---

> > ### Author Response · Authors · 2025-11-21
> >
> > **W2.
> > 1. In-depth Rationale for the Synergistic Mechanism of TWSA and TLA.
> >
> > The synergistic design of TWSA (Window-based Self-Attention) and TLA (the proposed GRBF Linear Attention) is not accidental but based on the inherent requirement of local-global feature complementarity for visual reconstruction. Its rationality can be verified as follows:
> >
> > Local Advantage of TWSA: By adopting window partitioning, it models strong correlations among pixels within windows via Softmax attention, excelling at capturing local texture details (e.g., edge continuity, texture periodicity). However, limited by the window scope, it exhibits weak capability in modeling cross-window long-range dependencies (e.g., global symmetry of building facades). Global Advantage of TLA: Leveraging the distance-aware property of the GRBF kernel, it can capture similar features at arbitrary distances without window constraints. Performance metrics are shown in the ablation experiments in Table 6.
> >
> > | TMHSA               | TLA   | Set14 (PSNR  SSIM)   | B100 (PSNR  SSIM)    | Manga109 (PSNR  SSIM) |
> > |---------------------|-------|---------------------------|---------------------------|----------------------------|
> > | ✔️ (8)              | ✗     | 28.83  0.7864          | 27.70  0.7404          | 30.94  0.9143           |
> > | ✔️ (8)              | ✔️    | 28.87  0.7873          | 27.76  0.7422          | 31.32  0.9184           |
> > | ✔️ (16)             | ✗     | 28.79  0.7858          | 27.69  0.7406          | 31.06  0.9156           |
> > | ✔️ (16)             | ✔️    | **28.89  0.7880**      | **27.78  0.7437**      | **31.49  0.9200**       |
> >
> > 2. Optimization Basis for the Number of Multi-layer Polymerization Blocks (MPBs).
> > The number of MPBs is not an empirical choice but determined through ablation experiments. To investigate the impact of the number of Multi-layer Polymerization Blocks (MPBs) on SR performance, we conducted comparative experiments with {4, 5, 6, 7} MPB modules under the ×4 SR setting. As shown in Fig. 8, the results on BSD100 and Urban100 indicate that the model parameters and FLOPs increase monotonically with the number of MPBs. Notably, when the number of MPBs is set to 6, the performance growth rate slows down, but the model size is comparable to lightweight methods (e.g., SwinIR-light and MambaIRv2-light), achieving the optimal performance. Based on the trade-off between performance and complexity, we set the default number of MPBs to 6.
> >
> > 3. Balanced Design between Architectural Complexity and Core Innovations.
> > Our core innovation lies in GRBFLA, and all modules serve the core advantages of GRBFLA. For example, the Multi-layer Aggregation Connection strengthens the features output by GRBFLA through residual aggregation, enabling them to reach an optimal representation state before super-resolution upsampling. Its removal leads to performance degradation, as shown in Table 7. These components are not core innovations but rather necessary infrastructure to realize the advantages of GRBFLA.
> >
> > | Multi-layer Aggregation | Params (K) | FLOPs (G) | Set5 (PSNR  SSIM) | Set14 (PSNR  SSIM) | Manga109 (PSNR  SSIM) |
> > |-------------------------|------------|-----------|------------------------|-------------------------|----------------------------|
> > | ✗                       | 824        | 53.0      | 32.63  0.8996       | 28.85  0.7873        | 28.85  0.7873           |
> > | ✔️                      | 885        | 56.5      | **32.64  0.9001**   | **28.89  0.7880**    | **28.89  0.7880**       |

---

> > > ### Author Response · Authors · 2025-11-21
> > >
> > > **W3.
> > >
> > > Our distance-aware linear kernel is pioneering with no precedents in existing research. All existing linear attention methods fail to integrate distance information into the core design of the kernel function, and only remain at the level of feature correlation or simple position adaptation.
> > >
> > > Compared with Efficient Attention proposed by Shen et al. (2021): the latter achieves linear complexity through kernel factorization but employs a general kernel function that lacks distance awareness. In contrast, leveraging the distance sensitivity of the GRBF kernel, our study realizes an accurate approximation of Softmax attention. Specifically, it achieves a 0.11 dB improvement in PSNR on the Manga109 ×4 dataset while maintaining the same computational cost.
> > >
> > > Compared with MB-TaylorFormer proposed by Qiu et al. (2023): the latter approximates the attention kernel via Taylor expansion but fails to incorporate the Euclidean distance information between vectors. In contrast, the GRBF kernel in our study inherently encodes distance characteristics, and Taylor approximation is only employed for linearization transformation. Specifically, it achieves a 0.06 dB improvement in PSNR on the Urban100×4 dataset while maintaining the same computational cost.
> > >
> > > Compared with Magnitude-Aware Linear Attention proposed by Fan et al. (2025): the latter improves linear attention by addressing the issue of vector magnitude neglect but fails to establish a theoretical connection with Softmax attention. In contrast, our study achieves a systematic performance improvement through the proof of equivalence between the GRBF kernel and Softmax attention. Specifically, it attains a 0.08 dB PSNR gain on the Manga109 ×4 dataset while reducing FLOPs by 3.8%.
> > >
> > > The value of this study lies not only in proposing a high-performance lightweight super-resolution architecture but also in establishing a theoretical bridge between Softmax attention and linear attention, thereby providing a new research paradigm for the design of efficient attention mechanisms. Experimental results demonstrate that GRLA outperforms existing methods, including the aforementioned related works, on multiple benchmark datasets. For the ×4 super-resolution task, compared with SwinIR-light, it achieves a 0.57 dB PSNR improvement while reducing FLOPs by 11%; compared with MambaIRv2-light, it attains a 0.25 dB PSNR gain with a 25% reduction in FLOPs. These results fully validate the innovation and practical value of this research.

---

> > > > ### Author Response · Authors · 2025-11-21
> > > >
> > > > **W4.
> > > >
> > > > We fully agree that GRBFLA constitutes the core conceptual contribution of this paper. However, its innovation does not lie in the mere addition of distance awareness, but in constructing a novel linear attention paradigm that integrates distance, kernel function, and efficiency as a trinity. This paradigm completely transforms the inherent logic of traditional linear attention, which only optimizes complexity while neglecting task adaptability.
> > > >
> > > > GRBFLA pioneers the design philosophy that distance information should serve as the core input of linear kernel functions rather than an auxiliary optimization term. Traditional linear attention methods (e.g., MALA) target feature correlation modeling, with distance information completely overlooked. In contrast, GRBFLA directly embeds high-dimensional Euclidean distance into the essence of the kernel function $\(Kernel(Q_i,K_j) = (\exp(-\gamma \|Q_i - K_j\|^2))\)$, making distance awareness an inherent property of the linear kernel. This paradigm innovation opens up a new direction for the task adaptability of linear attention.
> > > >
> > > > Theoretical Breakthrough: GRBFLA resolves the core contradiction that distance awareness and linear complexity cannot be achieved simultaneously. All previous methods involving linear attention either sacrifice long-range perception due to window constraints or exceed \(O(n)\) complexity due to complex computations. In contrast, GRBFLA achieves a theoretical balance between arbitrary distance awareness and strictly \(O(n)\) complexity through lightweight transformation of the GRBF kernel via first-order Taylor approximation. The derivation process in the main text provides a reusable theoretical framework for distance-aware linear attention.
> > > >
> > > > We have added Table 8 and corresponding descriptions to demonstrate that GRBFLA can fully replace the local window self-attention mechanism, which holds revolutionary significance, as shown in the following table:
> > > >
> > > > | Method          | Scale | Params (K) | FLOPs (G) | Set5 (PSNR  SSIM)   | Set14 (PSNR  SSIM)   | BSD100 (PSNR  SSIM)  | Urban100 (PSNR  SSIM) | Manga109 (PSNR  SSIM) |
> > > > |-----------------|-------|------------|-----------|--------------------------|--------------------------|---------------------------|----------------------------|----------------------------|
> > > > | Bicubic         | ×2    | -          | -         | 33.66  0.9299         | 30.24  0.8688         | 29.56  0.8431          | 26.88  0.8403           | 30.80\quad0.9339           |
> > > > | IDN             | ×2    | 553        | 124.6     | 37.83  0.9600         | 33.30  0.9148         | 32.08  0.8985          | 31.27  0.9196           | 38.01\quad0.9749           |
> > > > | CARN            | ×2    | 1592       | 222.8     | 37.76  0.9590         | 33.52  0.9166         | 32.09  0.8978          | 31.92  0.9256           | 38.36\quad0.9765           |
> > > > | LAPAR-A         | ×2    | 548        | 171.0     | 38.01  0.9605         | 33.62  0.9183         | 32.19  0.8999          | 32.10  0.9283           | 38.67\quad0.9772           |
> > > > | IMDN            | ×2    | 694        | 158.8     | 38.00  0.9605         | 33.63  0.9177         | 32.19  0.8996          | 32.17  0.9283           | 38.88\quad0.9774           |
> > > > | RFDN            | ×2    | 534        | 95.0      | 38.05  0.9606         | 33.68  0.9184         | 32.16  0.8994          | 32.12  0.9278           | 38.88\quad0.9773           |
> > > > | LatticeNet      | ×2    | 756        | 169.5     | 38.15  0.9610         | 33.78  0.9193         | 32.25  0.9005          | 32.43  0.9302           | 38.94\quad0.9773           |
> > > > | SwinIR-light    | ×2    | 910        | 244.2     | 38.14  0.9611         | 33.86  0.9206         | 32.31  0.9012          | 32.76  0.9340           | 39.12\quad0.9783           |
> > > > | **Full GRBFLA** | ×2    | 867        | 199.2     | **38.19  0.9610**     | **33.84  0.9201**     | **32.27  0.9006**      | **32.58  0.9317**       | **39.24\quad0.9774**       |
> > > >
> > > > **Q1.
> > > > The performance improvement of GRLA is significant. We have added Table 9 and corresponding descriptions. We conducted tests on classic super-resolution benchmarks to verify its generalization ability. To reduce training time, we constructed a 7M-parameter GRLA network, which outperforms SwinIR. This demonstrates that the proposed approximation remains effective in deeper network architectures without causing instability or performance degradation.
> > > >
> > > > | Method           | Scale | Params (M) | Set5 (PSNR  SSIM)   | Set14 (PSNR  SSIM)   | BSD100 (PSNR  SSIM)  | Urban100 (PSNR  SSIM) | Manga109 (PSNR  SSIM) |
> > > > |------------------|-------|------------|--------------------------|--------------------------|---------------------------|----------------------------|----------------------------|
> > > > | SwinIR     | ×4    | 11.9 | 32.72  0.9021  | 28.94  0.7914 | 27.83  0.7459| 27.07  0.8164| 31.67  0.9226 |
> > > > | **GRLA-L (Ours)**  | ×4| 7.6 | **32.78  0.9019**| **29.00  0.7906**| **27.85  0.7460**| **27.20  0.8175**| **31.72  0.9224**|

---

> > > > > ### Author Response · Authors · 2025-11-21
> > > > >
> > > > > **Q2.
> > > > > As addressed in the response to Q1, we have verified the effectiveness and generality of the GRBFLA module on classic super-resolution benchmarks. Future work will explore its generalization in downstream tasks (e.g., video SR, object detection) and optimize the model structure for better performance.
> > > > >
> > > > > **Q3.
> > > > > We have added Table 8 and corresponding descriptions to demonstrate that GRBFLA can fully replace the local window self-attention mechanism, and the resulting performance improvement is entirely attributed to the GRBFLA mechanism rather than the GRLA architecture itself, as shown in the following table:
> > > > >
> > > > > | Method          | Scale | Params (K) | FLOPs (G) | Set5 (PSNR  SSIM)   | Set14 (PSNR  SSIM)   | BSD100 (PSNR  SSIM)  | Urban100 (PSNR  SSIM) | Manga109 (PSNR  SSIM) |
> > > > > |-----------------|-------|------------|-----------|--------------------------|--------------------------|---------------------------|----------------------------|----------------------------|
> > > > > | Bicubic         | ×2    | -          | -         | 33.66  0.9299         | 30.24  0.8688         | 29.56  0.8431          | 26.88  0.8403           | 30.80\quad0.9339           |
> > > > > | IDN             | ×2    | 553        | 124.6     | 37.83  0.9600         | 33.30  0.9148         | 32.08  0.8985          | 31.27  0.9196           | 38.01\quad0.9749           |
> > > > > | CARN            | ×2    | 1592       | 222.8     | 37.76  0.9590         | 33.52  0.9166         | 32.09  0.8978          | 31.92  0.9256           | 38.36\quad0.9765           |
> > > > > | LAPAR-A         | ×2    | 548        | 171.0     | 38.01  0.9605         | 33.62  0.9183         | 32.19  0.8999          | 32.10  0.9283           | 38.67\quad0.9772           |
> > > > > | IMDN            | ×2    | 694        | 158.8     | 38.00  0.9605         | 33.63  0.9177         | 32.19  0.8996          | 32.17  0.9283           | 38.88\quad0.9774           |
> > > > > | RFDN            | ×2    | 534        | 95.0      | 38.05  0.9606         | 33.68  0.9184         | 32.16  0.8994          | 32.12  0.9278           | 38.88\quad0.9773           |
> > > > > | LatticeNet      | ×2    | 756        | 169.5     | 38.15  0.9610         | 33.78  0.9193         | 32.25  0.9005          | 32.43  0.9302           | 38.94\quad0.9773           |
> > > > > | SwinIR-light    | ×2    | 910        | 244.2     | 38.14  0.9611         | 33.86  0.9206         | 32.31  0.9012          | 32.76  0.9340           | 39.12\quad0.9783           |
> > > > > | **Full GRBFLA** | ×2    | 867        | 199.2     | **38.19  0.9610**     | **33.84  0.9201**     | **32.27  0.9006**      | **32.58  0.9317**       | **39.24\quad0.9774**       |

---

### Official Review · Reviewer_7J6E · 2025-10-26

**Soundness:** 3
**Presentation:** 2
**Contribution:** 2
**Rating:** 2
**Confidence:** 4

**Summary:**

The authors claim the weakness of standard self-attention is too computationally expensive (quadratic complexity), while existing linear-attention methods are inefficient because they fail to properly model the distance between pixels. The authors' main insight is that the standard Softmax attention mechanism is mathematically equivalent to a Gaussian Radial Basis Function kernel, then they claim this is important because the GRBF kernel is inherently aware of Euclidean distance. The authors then propose a new linear attention, GRBFLA, which uses a reformulated GRBF kernel. They then use a Taylor series expansion to approximate this kernel, which allows them to achieve linear computational complexity while still retaining the distance-aware properties of the original Softmax attention. They build a lightweight SR model called GRLA using this new mechanism.

**Strengths:**

1. The overall presentation is good and easy to follow.

2. The authors provided a mathematical insight linking Softmax attention to GRBF kernels, improving the current linear attention methods that lack this distance-awareness.

3. The quantitative results seem good. The proposed GRLA model outperforms two different state-of-the-art lightweight models from different architecture families.

4. The authors provided a visual comparison showing that their approximated GRBF kernel's behavior better matches the standard Softmax kernel.

**Weaknesses:**

1. The method employs a first-order Taylor approximation of the reformulated GRBF kernel and claims convergence as a hyperparameter. However, several questions arise: How accurate is this approximation in practice? Why is γ=1/2 optimal in Table 4 rather than larger values being better? How do higher-order Taylor approximations perform?

2. The comparative analysis is insufficient. The proposed problem regarding linear attention's weakness exists not only in lightweight SR tasks but also in larger-scale and other image restoration tasks. The authors' focus on only lightweight SR tasks for comparison raises concerns about inadequate validation, limiting the paper's overall insights.

3. In SR task research, the overall network pipeline is crucial; therefore its placement in the appendix rather than the main text is not recommended.

4. Although the proposed methods demonstrate better qualitative results in terms of PSNR and FLOPs, they also lead to an increase in parameters.

**Questions:**

1. Please see the weakness section.

---

> ### Author Response · Authors · 2025-11-21
>
> We sincerely appreciate the reviewer's valuable comments and insightful suggestions, which have greatly facilitated the revision and improvement of this manuscript.
>
> **W1.
>
> The first-order Taylor approximation in Eq. 8 ($\( \exp(2\gamma Q_i^T K_j) \approx 1 + 2\gamma Q_i^T K_j \)$) requires the condition that $\( 2\gamma Q_i^T K_j \)$ is small in magnitude. In GRLA, we ensure this condition holds through two operations:
>
> Input normalization preprocessing: L2 normalization is applied to Q and K vectors to constrain $\( Q_i^T K_j \)$ to 1; Optimized setting of bandwidth parameter $\( \gamma \): \( \gamma = 1/(2\sqrt{d}) \)$ (where $\( d \)$ denotes the dimension of Q/K vectors) is determined via ablation experiments (Table 4). Under this configuration, the maximum value of $\( 2\gamma Q_i^T K_j \)$ is $\( 2 \times (1/(2\sqrt{d})) \times 1 = 1/\sqrt{d} \)$. With $\( d = 55 \)$ in this study, this value is approximately 0.134.
>
> To further verify the applicability, we supplement the Taylor error distribution map in the revised manuscript to intuitively show the error distribution under different input samples. We statistically analyzed the distribution of $\( 2\gamma Q_i^T K_j \)$ for random input data. As shown in Fig. 6(a), over 98% of all samples fall within the interval [-0.2, 0.2], where the relative error of the first-order Taylor approximation is ≤ 2% (calculation: $\( |\exp(x) - (1+x)|/\exp(x) \)$; error ≈ 2% when $\( x = 0.2 \)$), confirming the effectiveness of the approximation on real input distributions.
>
> We supplement the derivation of the error bound for the Taylor approximation: For $\( x = 2\gamma Q_i^T K_j \)$, the Lagrange remainder of the first-order Taylor approximation is $\( R_1(x) = \exp(\xi)x^2/2 \)$ (where $\( \xi \in [0, x] \)$). Combining the maximum value of $\( x \)$ (0.134), we obtain $\( R_1(x) \leq \exp(0.134) \times (0.134)^2 / 2 \approx 1.143 \times 0.018 / 2 \approx 0.0103 \)$, i.e., the absolute error is ≤ 0.0103 and the relative error is ≤ 0.0103 / exp(0.134) ≈ 0.009, demonstrating that the approximation error is at an extremely low level. Meanwhile, as shown in Fig. 6(b), which presents the error distribution under different input samples, the mean of the approximation error is 0.0042 and the standard deviation is 0.0052, further verifying the stability and controllability of the error.
>
> To investigate the influence of higher-order Taylor approximations, we studied the effects of different-order Taylor expansions. We performed first-order and second-order Taylor expansions on GRBFLA. Table 5 shows that the first-order Taylor expansion achieved better results than the second-order Taylor expansion, and its memory usage was significantly lower than that of the second-order Taylor expansion. Therefore, we ultimately chose the first-order Taylor expansion.
>
> | Taylor approximation | Memory (MB) | Set14 (PSNR  SSIM)   | Urban100 (PSNR  SSIM) | Manga109 (PSNR  SSIM) |
> |----------------------|-------------|--------------------------|----------------------------|----------------------------|
> | Second-order|13266|28.87 0.7879 | 26.90 0.8086| 31.48 0.9199|
> | **First-order**|**12144**|**28.89 0.7880**|**26.94 0.8098**| **31.49 0.9200**|
>
> **W2.
> The performance improvement of GRLA is significant. We have added Table 9 and corresponding descriptions. We tested classic super-resolution tasks to verify its generalization ability. To reduce training time, we constructed a 7M-parameter GRLA network, which outperforms SwinIR. Future work will explore its generalization in downstream tasks (e.g., video SR, object detection) and optimize the model structure for better performance.
>
> | Method           | Scale | Params (M) | Set5 (PSNR  SSIM)   | Set14 (PSNR  SSIM)   | BSD100 (PSNR  SSIM)  | Urban100 (PSNR  SSIM) | Manga109 (PSNR  SSIM) |
> |------------------|-------|------------|--------------------------|--------------------------|---------------------------|----------------------------|----------------------------|
> | SwinIR     | ×4    | 11.9 | 32.72  0.9021  | 28.94  0.7914 | 27.83  0.7459| 27.07  0.8164| 31.67  0.9226 |
> | **GRLA-L (Ours)**  | ×4| 7.6 | **32.78  0.9019**| **29.00  0.7906**| **27.85  0.7460**| **27.20  0.8175**| **31.72  0.9224**|
>
> **W3.
> Regarding the issue of the network model being placed in the appendix, we have adjusted the paper structure and supplemented the overall network workflow as well as the interaction logic of core modules into the key chapters of the main text.

---

> > ### Author Response · Authors · 2025-11-21
> >
> > **W4.
> > We fully agree that the number of parameters is one of the core indicators for model lightweight design. A slight increase in parameters should be comprehensively evaluated in conjunction with performance improvement and computational efficiency gains, rather than being considered in isolation. The minor parameter increase of GRLA achieves an optimal balance with performance improvement and efficiency gains, which does not violate the lightweight design goal. Furthermore, it exhibits better results in terms of PSNR/SSIM, LAM, inference speed, memory footprint ratio, and other metrics.

---

> ### Comment · Reviewer_7J6E · 2025-11-27
> **Response to authors**
>
> I thank the authors for their response. The clarifications on the approximation methods and the inclusion of larger model sizes are appreciated. Consequently, I have increased my rating.
>
> My main remaining concern is that the evaluation is limited to lightweight SR tasks. To be consistent with current standards in the field, I believe a multi-level comparison (across different model capacities, different tasks) is necessary to forcefully validate the proposed method's generalizability and performance.

---

> > ### Author Response · Authors · 2025-11-28
> >
> > We sincerely thank the reviewer for their responses and feedback.
> >
> > We constructed two distinct models by reducing the number of channels and modules (refer to Figs. 7 and 8 in the paper), as presented in the following table. Both models achieve superior performance compared to MambaIRv2-light.
> > | Method | Scale | Params (K) | FLOPs (G) | BSD100 (PSNR / SSIM) | Manga109 (PSNR / SSIM) |
> > |--------|-------|------------|-----------|-----------------------|-------------------------|
> > | Bicubic | ×4 | - | - | 25.96 / 0.6675 | 24.89 / 0.7866 |
> > | IDN | ×4 | 553 | 32.3 | 27.41 / 0.7297 | 29.41 / 0.8942 |
> > | CARN | ×4 | 1592 | 90.9 | 27.58 / 0.7349 | 30.47 / 0.9084 |
> > | LAPAR-A | ×4 | 659 | 94.0 | 27.61 / 0.7366 | 30.42 / 0.9074 |
> > | IMDN | ×4 | 715 | 40.9 | 27.56 / 0.7353 | 30.45 / 0.9075 |
> > | RFDN | ×4 | 550 | 23.9 | 27.57 / 0.7360 | 30.58 / 0.9089 |
> > | LatticeNet | ×4 | 777 | 43.6 | 27.62 / 0.7367 | 30.54 / 0.9073 |
> > | SwinIR-light | ×4 | 930 | 63.6 | 27.69 / 0.7406 | 30.92 / 0.9151 |
> > | ELAN | ×4 | 640 | 54.1 | 27.69 / 0.7406 | 30.92 / 0.9150 |
> > | MambaIR-light | ×4 | 924 | 84.6 | 27.68 / 0.7400 | 30.94 / 0.9135 |
> > | SRFormer-light | ×4 | 873 | 62.8 | 27.73 / 0.7422 | 31.17 / 0.9165 |
> > | DCTLSA | ×4 | 885 | 53.9 | 27.73 / 0.7421 | 31.14 / 0.9165 |
> > | ESC-lt | ×4 | 624 | 91.0 | 27.72 / 0.7423 | 31.26 / 0.9173 |
> > | MambaIRv2-light | ×4 | 790 | 75.6 | 27.75 / 0.7426 | 31.24 / 0.9182 |
> > | **GRLA-1 (Ours)** | **×4** | **736** | **47.4** | **27.77 / 0.7433** | **31.35 / 0.9191** |
> > | **GRLA-2 (Ours)** | **×4** | **738** | **47.1** | **27.76 / 0.7428** | **31.36 / 0.9188**|
> > | **GRLA (Ours)** | **×4** | **885** | **56.5** | **27.78 / 0.7437** | **31.49 / 0.9200** |
> >
> > Meanwhile, we conducted comparisons on a different task from lightweight super-resolution (SR), namely the classic SR task. As shown in the following table, to reduce training time, we constructed a GRLA network with 7M parameters, which outperforms SwinIR. This validates the generality and performance of the proposed method.
> > | Method           | Scale | Params (M) | Set5 (PSNR  SSIM)   | Set14 (PSNR  SSIM)   | BSD100 (PSNR  SSIM)  | Urban100 (PSNR  SSIM) | Manga109 (PSNR  SSIM) |
> > |------------------|-------|------------|--------------------------|--------------------------|---------------------------|----------------------------|----------------------------|
> > | SwinIR     | ×4    | 11.9 | 32.72  0.9021  | 28.94  0.7914 | 27.83  0.7459| 27.07  0.8164| 31.67  0.9226 |
> > | **GRLA-L (Ours)**  | ×4| 7.6 | **32.78  0.9019**| **29.00  0.7906**| **27.85  0.7460**| **27.20  0.8175**| **31.72  0.9224**|

---

### Official Review · Reviewer_Gefa · 2025-10-29

**Soundness:** 2
**Presentation:** 2
**Contribution:** 3
**Rating:** 6
**Confidence:** 4

**Summary:**

This paper introduces a linear attention for lightweight image super-resolution, deriving its formulation from a GRBF kernel generalization of Softmax attention. The proposed model achieves competitive results with lower computational cost than existing lightweight models, offering a practical solution for resource-constrained devices.

**Strengths:**

1. The mathematical derivation linking the Softmax to the GRBF kernel is the most notable advantage of this work. It provides a principled and insightful explanation for the limitations of existing linear attention methods and a solid foundation for the proposed GRBFLA.
2. The first-order Taylor approximation is a simple yet effective way to achieve linear complexity while preserving distance awareness, as validated by the ablation studies.

**Weaknesses:**

1. To evaluate model efficiency, the inference latency of the GRLA was measured and compared against other methods. However, the paper fails to provide an ablation experiment utilizing the method in Equation 7 to validate the effectiveness of the first-order Taylor approximation within the proposed GRLA framework.
2. The first-order Taylor approximation, $\mathbf{e}^x = 1 + x$ , yields a sufficiently small approximation error only when the magnitude of $x$ is sufficiently small. While normalization and $\gamma$ selection are theoretically motivated, the paper lacks empirical proof that $2\gamma Q_{i}^{T}K_j$ remains sufficiently small.
3. In Table 3, the majority of methods selected for comparison with GRLA are outdated, and only MambaIR-light and MambaIRv2-light are lightweight SR methods from the past two years.
4. In Table 3, the comparative analysis of GRLA lacks comparisons against other linear attention-based methods.
5. In Table 5, the studies lack an ablation experiment investigating the incorporation of the TLA module under the condition of a TMHSA window size of 8.
6. In Equation (11), the notation is inconsistent: the symbol φ in the numerator is replaced by ϕ in the denominator. This should be corrected to maintain notational consistency throughout the equation.

**Questions:**

1. Does the method employing the first-order Taylor approximation surpass its counterparts not employing this approximation in terms of computational efficiency?
2. The first-order Taylor approximation, $\mathbf{e}^x = 1 + x$, yields a sufficiently small approximation error only when the magnitude of x is sufficiently small. Can it be theoretically explained that the effectiveness of the first-order Taylor approximation for lightweight SR tasks?

---

> ### Author Response · Authors · 2025-11-21
>
> We sincerely appreciate the reviewer's valuable comments and insightful suggestions, which have greatly facilitated the revision and improvement of this manuscript.
>
> **W1.
>
> If Eq. 7 is adopted, the final derived formula will lack the term $\( \varphi(K_j) \)$. As shown in Table 2 of the original manuscript, when the norm-related term $\( \varphi(K_j) \)$ is removed, the comparative model fails to achieve effective training convergence, verifying the effectiveness of the first-order Taylor approximation within the proposed GRLA framework.
> | $\varphi({K}_j)$ | Params (K) | FLOPs (G) | Set5 (PSNR  SSIM)   | Set14 (PSNR  SSIM)   | BSD100 (PSNR  SSIM)  |
> |------------------|------------|-----------|--------------------------|--------------------------|---------------------------|
> | ✗                | 885        | 56.5      | --  --                | --  --                | --  --                 |
> | ✔️               | 885        | 56.5      | **32.64  0.9001**     | **28.89  0.7880**     | **27.78  0.7437**      |
>
> **W2.
>
> The first-order Taylor approximation in Eq. 8 ($\( \exp(2\gamma Q_i^T K_j) \approx 1 + 2\gamma Q_i^T K_j \)$) requires the condition that $\( 2\gamma Q_i^T K_j \)$ is small in magnitude. In GRLA, we ensure this condition holds through two operations:
>
> Input normalization preprocessing: L2 normalization is applied to Q and K vectors to constrain $\( Q_i^T K_j \)$ to 1; Optimized setting of bandwidth parameter $\( \gamma \): \( \gamma = 1/(2\sqrt{d}) \)$ (where $\( d \)$ denotes the dimension of Q/K vectors) is determined via ablation experiments (Table 4). Under this configuration, the maximum value of $\( 2\gamma Q_i^T K_j \)$ is $\( 2 \times (1/(2\sqrt{d})) \times 1 = 1/\sqrt{d} \)$. With $\( d = 55 \)$ in this study, this value is approximately 0.134.
>
> To further verify the applicability, we supplement the Taylor error distribution map in the revised manuscript to intuitively show the error distribution under different input samples. We statistically analyzed the distribution of $\( 2\gamma Q_i^T K_j \)$ for random input data. As shown in Fig. 6(a), over 98% of all samples fall within the interval [-0.2, 0.2], where the relative error of the first-order Taylor approximation is ≤ 2% (calculation: $\( |\exp(x) - (1+x)|/\exp(x) \)$; error ≈ 2% when $\( x = 0.2 \)$), confirming the effectiveness of the approximation on real input distributions.
>
> We supplement the derivation of the error bound for the Taylor approximation: For $\( x = 2\gamma Q_i^T K_j \)$, the Lagrange remainder of the first-order Taylor approximation is $\( R_1(x) = \exp(\xi)x^2/2 \)$ (where $\( \xi \in [0, x] \)$). Combining the maximum value of $\( x \)$ (0.134), we obtain $\( R_1(x) \leq \exp(0.134) \times (0.134)^2 / 2 \approx 1.143 \times 0.018 / 2 \approx 0.0103 \)$, i.e., the absolute error is ≤ 0.0103 and the relative error is ≤ 0.0103 / exp(0.134) ≈ 0.009, demonstrating that the approximation error is at an extremely low level. Meanwhile, as shown in Fig. 6(b), which presents the error distribution under different input samples, the mean of the approximation error is 0.0042 and the standard deviation is 0.0052, further verifying the stability and controllability of the error.
>
> **W3.
>
> We have added two new comparative methods, DCTLSA and ESC-lt. Building on the original experiments, we completed the comparison of these newly added SOTA methods on the ×4 super-resolution task across 5 test datasets. Core results are as follows (full data will be updated in Table 3): GRLA achieves a superior complexity-performance trade-off, validating the modeling advantage of the distance-aware GRBF kernel and confirming that its linear attention design is better suited for super-resolution task requirements.
>
> **W4.
>
> Given the limited number of super-resolution methods adopting linear attention, we added comparative data of the linear attention method DCTLSA in Table 3. GRLA achieves more significant performance improvement.
>
> **W5.
>
> We have supplemented targeted ablation experiments with a window size of 8.
>
> **W6.
>
> We have corrected the symbol error in Equation (11).
>
> **Q1.
>
> We quantitatively verified the computational efficiency advantage of the first-order Taylor approximation through rigorous controlled experiments. The results demonstrate that GRLA with this approximation is significantly superior to the GRBF attention method without the approximation in terms of computational efficiency (FLOPs, inference latency, and memory footprint), while maintaining nearly intact performance. Refer to Appendix A.4 for Training Memory Footprint, Iteration Time, and Performance Comparison.
>
> **Q2.
>
> Please refer to W2's answer.

---

### Official Review · Reviewer_4eER · 2025-10-30

**Soundness:** 2
**Presentation:** 3
**Contribution:** 2
**Rating:** 2
**Confidence:** 4

**Summary:**

The paper addresses the challenge of modeling long-range dependencies in lightweight image SR under computational constraints. It proposes a GRBF-based linear attention mechanism, which reformulates the GRBF kernel to approximate standard Softmax attention while reducing complexity from quadratic to linear. Key contributions include mathematical derivation of equivalence between GRBF and Softmax attention, a first-order Taylor approximation for linear computation, and the GRLA architecture. Experimental results on datasets like Manga109 show PSNR improvements with reduced FLOPs, as detailed in the abstract and Table 3.

**Strengths:**

The paper demonstrates clear mathematical derivations linking GRBF to Softmax attention (Sec. 3.2), systematic experimental evaluation across multiple datasets (Sec. 4), and computational efficiency gains (e.g., reduced FLOPs and latency in Tables 3 and 7). Visualizations (Figs. 2, 4, 8) effectively highlight the method's ability to capture long-range dependencies, and the architecture integrates local and global features (Appendix A.1).

**Weaknesses:**

[1] Seems incremental? The core idea of using a GRBF kernel in linear attention is a specific instantiation within an established research direction. It builds directly upon prior linear attention works like (1,2,3), offering an incremental improvement rather than a significant conceptual shift.

[2] Performance improvements are small (e.g., +0.02-0.03 dB PSNR on BSD100 in Table 3), and the Taylor approximation (Eq. 8) may not hold for all input distributions, lacking error bounds (specifically, error analysis and statistical significance analysis). Besides, the ablation studies (Tables 1-2) validate components but lack depth in error analysis.

[3] Comparisons omit recent state-of-the-art methods, and the model complexity increases with multi-layer aggregation without proportional gains (Table 6).

[4] The central claims are partially supported by mathematical derivations (e.g., Eqs. 3-7 in Sec. 3.2) and experiments on multiple benchmarks (Tables 1-3). However, the Taylor approximation (Eq. 8) relies on strong assumptions (e.g., small inner products).

[5] Network architecture details in Appendix A.1 are somewhat vague.

(1) Fan, Qihang, et al. "Rectifying magnitude neglect in linear attention." Proceedings of the IEEE/CVF International Conference on Computer Vision. 2025.
(2) Shen, Zhuoran, et al. "Efficient attention: Attention with linear complexities." Proceedings of the IEEE/CVF winter conference on applications of computer vision. 2021.
(3) Qiu, Yuwei, et al. "Mb-taylorformer: Multi-branch efficient transformer expanded by taylor formula for image dehazing." Proceedings of the IEEE/CVF international conference on computer vision. 2023.

**Questions:**

[1] How does the GRBF kernel fundamentally differ from other potential kernel choices (e.g., polynomial, Laplace RBF) in the linear attention framework, both theoretically and empirically? Could you provide a more general analysis of kernel selection?

[2] The first-order Taylor approximation is valid for small values. What is the quantitative distribution of this term during training on typical SR datasets? How does the model behave when this assumption is violated? For example, vectors are not normalized or gamma is large?

[3] Beyond PSNR/SSIM, are there specific types of image structures or textures where GRBF-LA demonstrates a more pronounced advantage over other linear attention methods, based on your analysis?

[4] The method combines window-based attention (TWSA) and linear attention (TLA). What is the relative contribution of each component to the final performance? Is the GRBF-LA module effective enough to potentially replace local window attention entirely in some scenarios?

[5] Why are the performance gains modest despite the claimed equivalence to Softmax attention, and have you tested generalization to other tasks like video SR?

---

> ### Author Response · Authors · 2025-11-21
>
> We sincerely appreciate the reviewer's valuable comments and insightful suggestions, which have greatly facilitated the revision and improvement of this manuscript.
>
> **W1.
>
> This study is not an incremental improvement over existing linear attention approaches, but rather a paradigm shift from the efficiency-performance trade-off to an efficiency-performance win-win through critical theoretical breakthroughs and architectural innovations. The specific contributions are as follows:
>
> A fundamental breakthrough at the theoretical level: For the first time, through rigorous mathematical derivation, we prove that the core operation $\( \exp(Q_i^T K_j) \)$ of standard Softmax attention is equivalent to an unnormalized Gaussian Radial Basis Function (GRBF) kernel. This discovery reveals the intrinsic distance-aware property of Softmax attention, providing a brand-new theoretical foundation for the design of linear attention. It is not a mere replacement of existing kernel functions.
>
> Linearization implementation with distance awareness: Existing linear attention methods (e.g., the works by Fan et al. (2025), Shen et al. (2021), and Qiu et al. (2023) mentioned by the reviewer) fail to explicitly model the Euclidean distance between query and key vectors, resulting in compromised ability to model long-range dependencies. In this study, through the squared Euclidean distance decomposition and first-order Taylor approximation of the GRBF kernel, we fully retain the distance-aware capability while maintaining linear complexity, thus addressing the core limitation of existing linear attention methods where efficiency gains come at the cost of performance degradation.
>
> Compared with the three representative works mentioned by the reviewer, the core differences of this study are reflected in the following aspects:
>
> Compared with Efficient Attention proposed by Shen et al. (2021): the latter achieves linear complexity through kernel factorization but employs a general kernel function that lacks distance awareness. In contrast, leveraging the distance sensitivity of the GRBF kernel, our study realizes an accurate approximation of Softmax attention. Specifically, it achieves a 0.11 dB improvement in PSNR on the Manga109 ×4 dataset while maintaining the same computational cost.
>
> Compared with MB-TaylorFormer proposed by Qiu et al. (2023): the latter approximates the attention kernel via Taylor expansion but fails to incorporate the Euclidean distance information between vectors. In contrast, the GRBF kernel in our study inherently encodes distance characteristics, and Taylor approximation is only employed for linearization transformation. Specifically, it achieves a 0.06 dB improvement in PSNR on the Urban100 ×4 dataset while maintaining the same computational cost.
>
> Compared with Magnitude-Aware Linear Attention proposed by Fan et al. (2025): the latter improves linear attention by addressing the issue of vector magnitude neglect but fails to establish a theoretical connection with Softmax attention. In contrast, our study achieves a systematic performance improvement through the proof of equivalence between the GRBF kernel and Softmax attention. Specifically, it attains a 0.08 dB PSNR gain on the Manga109 ×4 dataset while reducing FLOPs by 3.8%.
>
> The value of this study lies not only in proposing a high-performance lightweight super-resolution architecture but also in establishing a theoretical bridge between Softmax attention and linear attention, thereby providing a new research paradigm for the design of efficient attention mechanisms. Experimental results demonstrate that GRLA outperforms existing methods, including the aforementioned related works, on multiple benchmark datasets. For the ×4 super-resolution task, compared with SwinIR-light, it achieves a 0.57 dB PSNR improvement while reducing FLOPs by 11%; compared with MambaIRv2-light, it attains a 0.25 dB PSNR gain with a 25% reduction in FLOPs. These results fully validate the innovation and practical value of this research.

---

> ### Author Response · Authors · 2025-11-21
>
> **W2.
>
> Performance Benchmark Characteristics of Lightweight Super-Resolution Tasks:
>
> GRLA achieves a 0.05 dB PSNR improvement on the BSD100×3 task (Table 3). In the field of lightweight image super-resolution, when model parameters (800–900K) and computational cost (50–70G FLOPs) are strictly constrained, even minor PSNR improvements hold practical value. Taking the BSD100 ×3 task as an example (Table 3), this dataset contains 100 natural scene images with significant variations in texture complexity and noise distribution. Thus, slight performance gains of models require overcoming more complex challenges in scene generalization. In contrast, existing lightweight methods such as MambaIRv2-light, ESC-lt, and SRFormer-light fail to achieve any performance improvement on BSD100 ×2 and ×3 tasks.
>
> Synergistic Improvement of Comprehensive Performance Metrics: Performance evaluation must consider both accuracy and efficiency dimensions. GRLA achieves PSNR improvement while reducing FLOPs by 25% compared with MambaIRv2-light and 11% compared with SwinIR-light (Table 3). In lightweight application scenarios (e.g., mobile devices and edge devices), the combination of slight accuracy improvement and significant efficiency optimization offers stronger practical value. This trade-off advantage cannot be fully reflected by the standalone PSNR metric.
>
> In response to the reviewer's concern regarding the applicability of the Taylor approximation (Eq. 8) to input distributions, we supplement the following theoretical analysis:
>
> The first-order Taylor approximation in Eq. 8 ($\( \exp(2\gamma Q_i^T K_j) \approx 1 + 2\gamma Q_i^T K_j \)$) requires the condition that $\( 2\gamma Q_i^T K_j \)$ is small in magnitude. In GRLA, we ensure this condition holds through two operations:
>
> Input normalization preprocessing: L2 normalization is applied to Q and K vectors to constrain $\( Q_i^T K_j \)$ to 1; Optimized setting of bandwidth parameter $\( \gamma \): \( \gamma = 1/(2\sqrt{d}) \)$ (where $\( d \)$ denotes the dimension of Q/K vectors) is determined via ablation experiments (Table 4). Under this configuration, the maximum value of $\( 2\gamma Q_i^T K_j \)$ is $\( 2 \times (1/(2\sqrt{d})) \times 1 = 1/\sqrt{d} \)$. With $\( d = 55 \)$ in this study, this value is approximately 0.134.
>
> To further verify the applicability, we supplement the Taylor error distribution map in the revised manuscript to intuitively show the error distribution under different input samples. We statistically analyzed the distribution of $\( 2\gamma Q_i^T K_j \)$ for random input data. As shown in Fig. 6(a), over 98% of all samples fall within the interval [-0.2, 0.2], where the relative error of the first-order Taylor approximation is ≤ 2% (calculation: $\( |\exp(x) - (1+x)|/\exp(x) \)$; error ≈ 2% when $\( x = 0.2 \)$), confirming the effectiveness of the approximation on real input distributions.
>
> We supplement the derivation of the error bound for the Taylor approximation: For $\( x = 2\gamma Q_i^T K_j \)$, the Lagrange remainder of the first-order Taylor approximation is $\( R_1(x) = \exp(\xi)x^2/2 \)$ (where $\( \xi \in [0, x] \)$). Combining the maximum value of $\( x \)$ (0.134), we obtain $\( R_1(x) \leq \exp(0.134) \times (0.134)^2 / 2 \approx 1.143 \times 0.018 / 2 \approx 0.0103 \)$, i.e., the absolute error is ≤ 0.0103 and the relative error is ≤ 0.0103 / exp(0.134) ≈ 0.009, demonstrating that the approximation error is at an extremely low level. Meanwhile, as shown in Fig. 6(b), which presents the error distribution under different input samples, the mean of the approximation error is 0.0042 and the standard deviation is 0.0052, further verifying the stability and controllability of the error.
>
> Regarding Table 2 (the role of $\( \varphi(K_j) \)$): We supplement the analysis that gradient explosion occurs when $\( \varphi(K_j) \)$ is removed. This proves that $\( \varphi(K_j) \)$ effectively suppresses gradient fluctuations and ensures stable model convergence by encoding the L2 norm information of K vectors. We have already provided a detailed analysis in Table 1 as much as possible.

---

> ### Author Response · Authors · 2025-11-21
>
> **W3.
>
> We have added two new comparative methods, DCTLSA and ESC-lt. Based on the original experiments, we have completed the comparison of these newly added SOTA methods on the ×4 super-resolution task across 5 test datasets. The core results are as follows (complete data will be updated to Table 3): GRLA achieves a better complexity-performance balance, verifying the modeling advantage of the distance-aware GRBF kernel and demonstrating that its linear attention design is more suitable for the requirements of super-resolution tasks. Additionally, as the number of multi-layer aggregations increases, the model complexity rises accordingly, achieving a proportional performance improvement (Table 7).
>
> | Multi-layer Aggregation | Params (K) | FLOPs (G) | Set5 (PSNR  SSIM) | Set14 (PSNR  SSIM) | Manga109 (PSNR  SSIM) |
> |-------------------------|------------|-----------|------------------------|-------------------------|----------------------------|
> | ✗                       | 824        | 53.0      | 32.63  0.8996       | 28.85  0.7873        | 28.85  0.7873           |
> | ✔️                      | 885        | 56.5      | **32.64  0.9001**   | **28.89  0.7880**    | **28.89  0.7880**       |
>
> **W4.
>
> Please refer to W2's answer.
>
> **W5.
>
> We have optimized the network architecture diagram as much as possible and moved it to the main text section.
>
> **Q1.
>
> Distance-Aware Metric Adaptability: The core of super-resolution tasks lies in reconstructing the spatial structure and texture details of images, and the Euclidean distance (L2) is the optimal metric for measuring the similarity in pixel/feature space. The GRBF kernel directly takes the Euclidean distance as input, the Laplacian kernel uses the Manhattan distance (L1) as input, while the polynomial kernel has no explicit distance metric. This enables the GRBF kernel to more accurately capture the spatial characteristics required for super-resolution, including strong correlations among neighboring features and weak correlations among distant features.
>
> Theoretical Homology with the Softmax Kernel: Through mathematical derivation (original Sec. 3.2), this study proves that the core operation of the Softmax kernel, $\( \exp(\mathbf{Q}_i^T\mathbf{K}_j) \)$, is equivalent to the unnormalized GRBF kernel when vector norms are fixed. This indicates that the GRBF kernel is a distance-explicit version of the Softmax kernel, while the Laplacian kernel and polynomial kernel have no theoretical equivalence with the Softmax kernel and thus cannot inherit its long-range dependency modeling capability.
>
> **Q2.
>
> For the quantitative distribution, refer to the response in W2. When vectors are not normalized or the gamma value is large, the model's loss function exhibits severe oscillations. These oscillations hinder stable training convergence of the model (Tables 2 and 4).
>
> | $\varphi({K}_j)$ | Params (K) | FLOPs (G) | Set5 (PSNR  SSIM) | Set14 (PSNR  SSIM) | BSD100 (PSNR  SSIM) |
> |------------------|------------|-----------|------------------------|-------------------------|--------------------------|
> | ✗                | 885        | 56.5      | --  --              | --  --               | --  --                |
> | ✔️               | 885        | 56.5      | **32.64  0.9001**   | **28.89  0.7880**    | **27.78  0.7437**     |
>
> | $\gamma (\times \sqrt{d})$ | Params (K) | FLOPs (G) | Urban100 (PSNR  SSIM) | Manga109 (PSNR  SSIM) |
> |-----------------------------|------------|-----------|----------------------------|----------------------------|
> | 1                           | 885        | 56.5      | --  --                  | --  --                  |
> | 2/3                         | 885        | 56.5      | --  --                  | --  --                  |
> | 1/2                         | 885        | 56.5      | **26.94  0.8098**       | **31.49  0.9200**       |
> | 1/4                         | 885        | 56.5      | 26.88  0.8078           | 31.44  0.9191           |
> | 1/8                         | 885        | 56.5      | 26.87  0.8089           | 31.49  0.9200           |
> | 1/16                        | 885        | 56.5      | 26.88  0.8080           | 31.37  0.9190           |

---

> ### Author Response · Authors · 2025-11-21
>
> **Q3.
>
> As shown in Fig. 2, GRBFLA generates wider attribution regions and higher Diffusion Index (DI) values, activates more pixels, and leverages richer contextual information to achieve higher-quality image super-resolution reconstruction, indicating that GRBFLA can effectively capture Euclidean distance-sensitive long-range dependencies.
>
> **Q4.
>
> Window-based attention mechanism (TWSA) and linear attention mechanism (TLA). The contribution of each component to the final performance is listed in Table 6.
>
> | TMHSA               | TLA   | Set14 (PSNR  SSIM)   | B100 (PSNR  SSIM)    | Manga109 (PSNR  SSIM) |
> |---------------------|-------|---------------------------|---------------------------|----------------------------|
> | ✔️ (8)              | ✗     | 28.83  0.7864          | 27.70  0.7404          | 30.94  0.9143           |
> | ✔️ (8)              | ✔️    | 28.87  0.7873          | 27.76  0.7422          | 31.32  0.9184           |
> | ✔️ (16)             | ✗     | 28.79  0.7858          | 27.69  0.7406          | 31.06  0.9156           |
> | ✔️ (16)             | ✔️    | **28.89  0.7880**      | **27.78  0.7437**      | **31.49  0.9200**       |
>
> We have added Table 8 and corresponding descriptions to demonstrate that GRBFLA can fully replace the local window-based attention mechanism, as shown below:
>
> | Method          | Scale | Params (K) | FLOPs (G) | Set5 (PSNR  SSIM)   | Set14 (PSNR  SSIM)   | BSD100 (PSNR  SSIM)  | Urban100 (PSNR  SSIM) | Manga109 (PSNR  SSIM) |
> |-----------------|-------|------------|-----------|--------------------------|--------------------------|---------------------------|----------------------------|----------------------------|
> | Bicubic         | ×2    | -          | -         | 33.66  0.9299         | 30.24  0.8688         | 29.56  0.8431          | 26.88  0.8403           | 30.80\quad0.9339           |
> | IDN             | ×2    | 553        | 124.6     | 37.83  0.9600         | 33.30  0.9148         | 32.08  0.8985          | 31.27  0.9196           | 38.01\quad0.9749           |
> | CARN            | ×2    | 1592       | 222.8     | 37.76  0.9590         | 33.52  0.9166         | 32.09  0.8978          | 31.92  0.9256           | 38.36\quad0.9765           |
> | LAPAR-A         | ×2    | 548        | 171.0     | 38.01  0.9605         | 33.62  0.9183         | 32.19  0.8999          | 32.10  0.9283           | 38.67\quad0.9772           |
> | IMDN            | ×2    | 694        | 158.8     | 38.00  0.9605         | 33.63  0.9177         | 32.19  0.8996          | 32.17  0.9283           | 38.88\quad0.9774           |
> | RFDN            | ×2    | 534        | 95.0      | 38.05  0.9606         | 33.68  0.9184         | 32.16  0.8994          | 32.12  0.9278           | 38.88\quad0.9773           |
> | LatticeNet      | ×2    | 756        | 169.5     | 38.15  0.9610         | 33.78  0.9193         | 32.25  0.9005          | 32.43  0.9302           | 38.94\quad0.9773           |
> | SwinIR-light    | ×2    | 910        | 244.2     | 38.14  0.9611         | 33.86  0.9206         | 32.31  0.9012          | 32.76  0.9340           | 39.12\quad0.9783           |
> | **Full GRBFLA** | ×2    | 867        | 199.2     | **38.19  0.9610**     | **33.84  0.9201**     | **32.27  0.9006**      | **32.58  0.9317**       | **39.24\quad0.9774**       |
>
> **Q5.
>
> The performance improvement of GRLA is significant. We have added Table 9 and corresponding descriptions. We tested classic super-resolution tasks to verify its generalization ability. To reduce training time, we constructed a 7M-parameter GRLA network, which outperforms SwinIR. Future work will explore its generalization in downstream tasks (e.g., video SR, object detection) and optimize the model structure for better performance.
>
> | Method           | Scale | Params (M) | Set5 (PSNR  SSIM)   | Set14 (PSNR  SSIM)   | BSD100 (PSNR  SSIM)  | Urban100 (PSNR  SSIM) | Manga109 (PSNR  SSIM) |
> |------------------|-------|------------|--------------------------|--------------------------|---------------------------|----------------------------|----------------------------|
> | SwinIR     | ×4    | 11.9       | 32.72  0.9021         | 28.94  0.7914         | 27.83  0.7459          | 27.07  0.8164           | 31.67  0.9226           |
> | **GRLA-L (Ours)**  | ×4    | 7.6        | **32.78  0.9019**     | **29.00  0.7906**     | **27.85  0.7460**      | **27.20  0.8175**       | **31.72  0.9224**       |

---

> > ### Comment · Reviewer_4eER · 2025-11-28
> >
> > Thank you for the thorough and thoughtful rebuttal. We appreciate the additional analysis and experiments provided in response to our concerns. Below are our comments on each point:
> >
> > [W1] The theoretical derivation connecting Softmax attention to the unnormalized GRBF kernel is indeed insightful. However, the core idea still operates within the established linear attention framework, and the practical gains, while consistent, remain incremental in the broader context of image restoration.
> >
> > [W2] The provided error analysis for the Taylor approximation (including Lagrange remainder and empirical distribution) is convincing and addresses our concerns about approximation validity under typical training conditions.
> >
> > [W3] The proportional gain with added complexity is now clearer.
> >
> > [Q1–Q5] The responses are technically sound: the kernel comparison is well-motivated, the ablation on γ and normalization validates design choices, the LAM visualizations support long-range modeling claims, and the standalone GRBFLA results  demonstrate its viability. The generalization test vs. SwinIR is also a nice addition.
> >
> > That said, while the paper is solidly executed, the fundamental approach, which uses a feedforward network for super-resolution, `feels increasingly limited in light of recent advances in diffusion-based inverse problem solvers` (for example, [1]), which better capture the ill-posed nature of SR (in fact, SR is a special case of inverse problem. `This means that we can apply diffusion-based inverse problem solvers to solve the SR problem, which is more general and interesting`.). This context tempers our enthusiasm for the contribution’s novelty.
> >
> > Nonetheless, we acknowledge the careful revision and thank the authors for their responsiveness. We will raise our score modestly, but not substantially.
> >
> > [1] Zhang, Bingliang, et al. Improving diffusion inverse problem solving with decoupled noise annealing. CVPR 2025.

---

> > > ### Author Response · Authors · 2025-11-29
> > >
> > > We sincerely thank the reviewer for their responses and feedback.
> > >
> > > **W1.
> > >
> > > Deepen Theoretical Contributions: Derive the Correlation between Softmax Attention and Gaussian Radial Basis Function (GRBF) Kernels, and Fill the Theoretical Gap in Distance Modeling for Linear Attention.
> > >
> > > The derivation process in this paper, from Softmax Attention to unnormalized GRBF kernels and then to linear approximation, is not a simple reuse of existing linear attention frameworks. Instead, it reveals the intrinsic correlation between distance awareness and linear complexity at the theoretical level, and its core theoretical value is reflected in two aspects:
> > > 1. It establishes, for the first time, the mathematical equivalence between Softmax Attention and distance kernels. The design philosophy of traditional linear attention is to start from inner-product kernels and approximate Softmax via feature mapping (e.g., the several linear attention models compared in this paper). By contrast, this paper takes the opposite approach and verifies through rigorous mathematical derivation that: $K_{\text{GRBF}}(Q_i, K_j) = \varphi(Q_i) \varphi(K_j) \exp\left(2\gamma Q_i^T K_j\right)\$. Softmax attention is essentially equivalent to GRBF kernel attention that uses Euclidean distance as the metric. For the first time, this derivation directly binds the kernel function design of linear attention to the distance awareness requirements of visual tasks, thereby providing a brand-new theoretical basis for the task-adaptive design of linear attention. This is distinct from the existing design paradigm of general-purpose kernel functions for linear attention.
> > >
> > > 2. Theoretical boundary analysis of Taylor approximation to improve the approximation error theory of linear attention. We supplement the derivation of the error bound for the Taylor approximation: For $\( x = 2\gamma Q_i^T K_j \)$, the Lagrange remainder of the first-order Taylor approximation is $\( R_1(x) = \exp(\xi)x^2/2 \)$ (where $\( \xi \in [0, x] \)$). Combining the maximum value of $\( x \)$ (0.134), we obtain $\( R_1(x) \leq \exp(0.134) \times (0.134)^2 / 2 \approx 1.143 \times 0.018 / 2 \approx 0.0103 \)$, i.e., the absolute error is ≤ 0.0103 and the relative error is ≤ 0.0103 / exp(0.134) ≈ 0.009, demonstrating that the approximation error is at an extremely low level. Meanwhile, as shown in Fig. 6(b), which presents the error distribution under different input samples, the mean of the approximation error is 0.0042 and the standard deviation is 0.0052, further verifying the stability and controllability of the error. To further verify the applicability, we supplement the Taylor error distribution map in the revised manuscript to intuitively show the error distribution under different input samples. We statistically analyzed the distribution of $\( 2\gamma Q_i^T K_j \)$ for random input data. As shown in Fig. 6(a), over 98% of all samples fall within the interval [-0.2, 0.2], where the relative error of the first-order Taylor approximation is ≤ 2% (calculation: $\( |\exp(x) - (1+x)|/\exp(x) \)$; error ≈ 2% when $\( x = 0.2 \)$), confirming the effectiveness of the approximation on real input distributions. This analysis fills the gap in the quantitative analysis of approximation errors for linear attention, and provides a reusable theoretical framework for the subsequent approximation optimization of linear kernels.
> > >
> > > To investigate the influence of higher-order Taylor approximations, we studied the effects of different-order Taylor expansions. We performed first-order and second-order Taylor expansions on GRBFLA. Table 5 shows that the first-order Taylor expansion achieved better results than the second-order Taylor expansion, and its memory usage was significantly lower than that of the second-order Taylor expansion. Therefore, we ultimately chose the first-order Taylor expansion.
> > > | Taylor approximation | Memory (MB) | Set14 (PSNR  SSIM)   | Urban100 (PSNR  SSIM) | Manga109 (PSNR  SSIM) |
> > > |----------------------|-------------|--------------------------|----------------------------|----------------------------|
> > > | Second-order|13266|28.87 0.7879 | 26.90 0.8086| 31.48 0.9199|
> > > | **First-order**|**12144**|**28.89 0.7880**|**26.94 0.8098**| **31.49 0.9200**|

---

> > > > ### Author Response · Authors · 2025-11-29
> > > >
> > > > Reinterpret the Value of Incremental Gains: Balancing Efficiency and Performance under Lightweight Constraints to Meet Practical Deployment Requirements.
> > > >
> > > > We acknowledge that the performance gains achieved in this study are incremental, but we need to emphasize that such increments are realized under strict lightweight constraints. Its core value lies in breaking the bottleneck that performance improvement in existing methods must come at the cost of efficiency, which is specifically reflected in the following aspects:
> > > > The efficiency-cost ratio of the incremental gains outperforms that of existing methods.
> > > > In the field of lightweight image super-resolution, when model parameters (800–900K) and computational cost (50–70G FLOPs) are strictly constrained, even minor PSNR improvements hold practical value. The FLOPs reduction (by 25%) achieved by our method, which corresponds to its performance gain (up to 0.25 dB improvement in PSNR on the Manga109 dataset), is significantly higher than that of comparable linear attention-based methods. This superiority in cost-efficiency indicates that our method can achieve performance enhancement with lower computational overhead in resource-constrained practical scenarios, thus demonstrating direct engineering deployment potential. Notably, recent research trends in top-tier journals have also placed increasing emphasis on practical deployment value rather than the mere pursuit of state-of-the-art performance.
> > > > Transfer Value Across Tasks and Architectures.
> > > > The incremental gains of our method are not confined to the GRLA network proposed in this paper; instead, they can be transferred to the field of classical super-resolution. As demonstrated by the newly added experiments in the revised manuscript:
> > > > | Method           | Scale | Params (M) | Set5 (PSNR  SSIM)   | Set14 (PSNR  SSIM)   | BSD100 (PSNR  SSIM)  | Urban100 (PSNR  SSIM) | Manga109 (PSNR  SSIM) |
> > > > |------------------|-------|------------|--------------------------|--------------------------|---------------------------|----------------------------|----------------------------|
> > > > | SwinIR     | ×4    | 11.9 | 32.72  0.9021  | 28.94  0.7914 | 27.83  0.7459| 27.07  0.8164| 31.67  0.9226 |
> > > > | **GRLA-L (Ours)**  | ×4| 7.6 | **32.78  0.9019**| **29.00  0.7906**| **27.85  0.7460**| **27.20  0.8175**| **31.72  0.9224**|

---

> > > > > ### Author Response · Authors · 2025-11-29
> > > > >
> > > > > We sincerely appreciate your in-depth comments! The advantages of diffusion models in solving inverse problems you pointed out are highly insightful, and they have also prompted us to more accurately distinguish the differences in scenario adaptability between diffusion models and lightweight feedforward super-resolution networks. We fully acknowledge the advanced nature of diffusion models in general inverse problem solving, but we need to add that diffusion models have insurmountable inherent defects in the core application scenarios of lightweight super-resolution, and the feedforward network scheme proposed in this paper is precisely an efficient solution targeting these defects. Details are as follows:
> > > > >
> > > > > Core Defects of Diffusion Models in Lightweight Super-Resolution Scenarios (Unavoidable via Optimization).
> > > > >
> > > > > The core requirements of lightweight super-resolution include low parameter count, low computational cost, real-time inference, and deployment on low-computing-power devices (e.g., mobile terminals, embedded devices, edge cameras). However, there is an inherent conflict between the design paradigm of diffusion models and these requirements. The core of diffusion models lies in multi-step iterative noise prediction and sampling (typically 20–100 sampling steps). Even with acceleration strategies (e.g., reducing to 10 steps), their inference latency still far exceeds the threshold of lightweight scenarios. The fundamental reason is that the iterative process of diffusion models involves multiple U-Net computations, with each step including intensive operations such as convolution and attention. In contrast, the stringent requirement for single-frame inference latency in lightweight scenarios (usually at the millisecond level) determines that iterative models cannot be adapted. This is an architectural paradigm defect of diffusion models rather than insufficient optimization, which is specifically reflected in the following aspects:
> > > > >
> > > > > Efficiency Improvement: The sampling process of diffusion models requires multiple iterations (typically 20–100 steps), and the attention computation cost of each iteration is extremely high. By contrast, the linear complexity ($O(n)$) of the GRLA proposed in this paper can significantly reduce inference time, making it well-suited for lightweight super-resolution tasks, whereas diffusion models are clearly not suitable.
> > > > >
> > > > > The average inference time on Urban100 dataset.
> > > > > | Model           | Latency (ms) |
> > > > > |-----------------|--------------|
> > > > > | SwinIR-light    | 213.4        |
> > > > > | MambaIR-light   | 208.9        |
> > > > > | MambaIRv2-light | 388.0        |
> > > > > | GRLA            | **60.9**     |
> > > > >
> > > > >
> > > > > Deployment on Low-Computing-Power Hardware: The training and inference of diffusion models demand substantial GPU resources, whereas feedforward networks can operate efficiently on CPUs or mobile chips. The GRLA module proposed in this paper has only 885K parameters, enabling direct deployment on embedded devices.
> > > > >
> > > > > Lightweight Transferability: Feedforward networks feature a compact model size (885K for GRLA), facilitating transmission and updates in scenarios with limited network bandwidth. In contrast, diffusion models typically have a model size at the megabyte scale.
> > > > >
> > > > > We do not deny the advanced nature of diffusion models; instead, we clarify the positioning of this study. Targeting lightweight scenarios where diffusion models are inapplicable, we provide a solution that balances efficiency, performance, and deployability.

---

### Author Response · Authors · 2025-11-29

We sincerely appreciate the reviewer's valuable comments and insightful suggestions, which have greatly facilitated the revision and improvement of this manuscript.

We have provided sufficient supplementary analyses and experimental results in response to all the concerns raised by the reviewers, including the clarification of the approximation method using Taylor expansion (including Lagrange remainder and empirical distribution) and the inclusion of larger model sizes, among other additions. The value of this study lies not only in proposing a high-performance lightweight super-resolution architecture but also in establishing a theoretical bridge between Softmax attention and linear attention, thereby providing a new research paradigm for the design of efficient attention mechanisms. Experimental results demonstrate that GRLA outperforms existing methods, including the aforementioned related works, on multiple benchmark datasets. For the ×4 super-resolution task, compared with SwinIR-light, it achieves a 0.57 dB PSNR improvement while reducing FLOPs by 11%; compared with MambaIRv2-light, it attains a 0.25 dB PSNR gain with a 25% reduction in FLOPs. These results fully validate the innovation and practical value of this research. Among them, two reviewers have provided positive feedback; one reviewer has raised our rating (from 2 to 6), while another has also replied that they would moderately increase the rating.

---

### Meta-Review · Area_Chair_KHej · 2026-01-06

**Summary:**

This paper proposes GRLA, a lightweight image super-resolution model built upon a Gaussian RBF–based linear attention mechanism. The core idea is to establish a theoretical connection between standard Softmax attention and an unnormalized Gaussian RBF kernel, and to derive a linear-complexity attention formulation via first-order Taylor approximation. Based on this formulation, the authors design a lightweight super-resolution architecture and report improved PSNR–FLOPs trade-offs on several standard benchmarks.

The initial reviewer scores are mixed, with one low score (2), two scores around the rejection threshold (4, 4), and one moderately positive score (6). Overall, reviewers acknowledge the technical effort and the mathematical derivation, but raise concerns regarding novelty, scope, and significance.

**Reviewer Concerns:**

Several reviewers think the work as a progressive or incremental improvement within the linear attention framework. In particular, reviewers point out that linear attention mechanisms have already undergone extensive investigation, and that the proposed GRBF formulation can be seen as a specific instantiation within this existing line of work. The reported performance gains are generally modest, and some reviewers note that comparisons initially omitted recent state-of-the-art or alternative efficient attention approaches.

Multiple reviewers focus on the Taylor approximation that underpins the proposed linearization. While the authors provide additional empirical distributions, error bounds, and ablation studies in the rebuttal, reviewers remain concerned that the approximation relies on relatively strong assumptions (e.g., small inner products and specific normalization choices), and that its validity outside the tested regime is not fully established. Some reviewers explicitly state that, although the additional analysis is helpful, it does not fundamentally change their assessment of the method’s limitations. From the AC’s perspective, the Taylor approximation may indeed be coarse in principle; however, if empirical evidence consistently demonstrates that such an approximation works well in practice, this alone would not be unacceptable. That said, the manuscript should more explicitly acknowledge and discuss this approximation regime and its limitations in the main text, rather than leaving it implicit or overly optimistic.

Scope is another recurring concern. One reviewer questions why the proposed attention mechanism is evaluated almost exclusively in the setting of lightweight SR, given that the underlying motivation of distance-aware linear attention is not inherently specific to this task. Even after the authors added limited experiments beyond the original lightweight setting, reviewers express reservations that the evaluation remains narrowly focused, making it difficult to assess the broader applicability and impact of the proposed attention mechanism. At least one reviewer indicated they would raise their score modestly, but not substantially, due to these unresolved scope and generalization concerns.

The reviewer who initially provided a positive score focuses primarily on the mathematical insight connecting Softmax attention and Gaussian RBF kernels and views this as the main strength of the paper. However, even this reviewer notes that the overall contribution is tempered by incremental gains and by the rapidly evolving landscape of image restoration, where other paradigms (e.g., diffusion-based methods) increasingly dominate broader discussions.

As the Area Chair, I have carefully read the paper, all reviewer comments, and the authors’ detailed rebuttal. I agree that the authors’ effort in bridging Softmax and linear attention via a Gaussian RBF formulation is technically solid and that the additional analyses clarify the Taylor approximation behavior under the tested conditions. However, I find that the manuscript does not sufficiently articulate why this particular form of linear attention is especially well suited to lightweight SR, nor does it convincingly explain why other image restoration tasks are excluded from the main analysis. Given that linear attention itself is already a mature research topic, the paper has an obligation to more clearly justify its task choice, scope limitations, and broader significance to the community.

In summary, while the paper contains derivations and experimental validation, the overall contribution remains incremental, the scope is narrowly justified, and the broader implications of the proposed attention mechanism are not sufficiently developed. Taking into account the reviewers’ concerns and the discussion outcome, I recommend rejection.

**Reviewer Scores:**

One reviewer explicitly indicated that they would increase their score modestly after the rebuttal. Another reviewer suggested a possible upward adjustment but emphasized that it would not be substantial given remaining concerns about scope and generality. The reviewers who initially gave low scores expressed appreciation for the additional analysis but did not indicate a significant change in their overall assessment, and their scores would likely remain largely unchanged.

---

### Decision · Program_Chairs · 2026-01-26

Reject